# NEURAL EVOLUTIONARY KERNEL METHOD: A KNOWLEDGE-BASED LEARNING ARCHITECHTURE FOR EVOLUTIONARY PDES

## ABSTRACT

Numerical solution of partial differential equations (PDEs) plays a vital role in various fields of science and engineering. In recent years, deep neural networks (DNNs) have emerged as a powerful tool for solving PDEs. DNN-based methods exploit the approximation capabilities of neural networks to obtain solutions to PDEs in general domains or high-dimensional spaces. However, many of these methods lack the use of mathematical prior knowledge, and DNN-based methods usually require a large number of sample points and parameters, making them computationally expensive and challenging to train. This paper aims to introduce a novel method named the Neural Evolutionary Kernel Method (NEKM) for solving a class of evolutionary PDEs through DNNs based kernels. By using operator splitting and boundary integral techniques, we propose particular neural network architectures which approximate evolutionary kernels of solutions and preserve structures of time-dependent PDEs. Mathematical prior knowledge are naturally built into these DNNs based kernels through convolutional representation with pre-trained Green functions, leading to serious reduction in the number of parameters in the NEKM and very efficient training processes. Experimental results demonstrate the efficiency and accuracy of the NEKM in solving heat equations and Allen-Cahn equations in complex domains and on manifolds, showcasing its promising potential for applications in data driven scientific computing.

## 1 INTRODUCTION

Partial differential equations (PDEs) find extensive applications in describing a wide range of physical and chemical phenomena, including but not limited to diffusion, heat transfer, fluid dynamics, and other related processes. For general PDEs, obtaining exact solutions is usually not feasible, while numerical approximations are often important alternatives for scientific discovery. Over the past century, numerous numerical methods based on linear approximation theory have been developed to solve PDEs, including finite difference/finite element/finite volume methods, and spectral method. However, these traditional methods usually suffer from the curse of dimensionality (E et al., 2019). And establishing high-order and sufficiently stable numerical schemes are often difficult for problems in complex geometries.

With the Universal Approximation Theorem of neural network by Cybenko (1989) as the theoretical support, the ability of neural network in approximating function has been widely explored (E et al., 2019; Barron, 1993; Shen et al., 2022; Zhang et al., 2022). As an alternative, deep neural networks (DNNs)-based methods for solving PDEs have emerged as a promising approach to solving PDEs. Many of these methods can bypass the need for explicit discretization of PDEs and learn maps between the specific spaces which are helpful for finding solutions of PDEs. Numerous studies have focused on the development of DNN-based approaches for solving a wide range of PDEs, which will be discussed in detail in section 1.1.

When utilizing DNN-based methods for solving PDEs, there are several frequently encountered limitations. Many of these methods necessitate sampling in a large domain covering feature space of the equation, and achieving better solutions requires gathering more comprehensive information from a large number of sample points. Furthermore, the selection of network architecture and optimizer can

also impact the convergence of training process. In extremal cases, when we are dealing with PDEs with oscillatory or even singular solutions, many of these DNN-based methods can even encounter failure because the appearance of high-order derivatives may lead to instability in training (Yang & Zhu, 2021; Lyu et al., 2022).

This paper presents the Neural Evolutionary Kernel Method (NEKM), a novel approach for solving a class of evolutionary PDEs. The proposed method employs the idea of operator splitting to divide numerical solution of a semi-linear PDE into two alternating steps: analytically (or numerically) solving the corresponding nonlinear ordinary differential equation (ODE) to obtain a flow map and numerically integrating the related linear PDE using a convolution kernel. This natually gives rise to a convolution block with an appropriate activation function. By utilizing fundamental solution of differential operator and boundary integral representation of solution in the given geometry, we recast the approximation of the convolution block into training of neural networks with squared loss over the boundary of the domain. The use of mathematical prior knowledge (such as fundamental solutions and geometry information in boundary integrals) facilitates the approximation capability of our proposed networks, leading to serious reduction in the number of training parameters. Moreover, boundary integral representations only require sampling from the boundary that is one dimension lower than the whole domain, which improves training efficiency. The proposed method provides a generic neural network framework for solution to a class of semi-linear PDEs, which is applicable to problems in complex domains.

## 1.1 RELATED WORKS

The utilization of neural networks for solving PDEs has garnered substantial interest in recent years owing to its capability to provide precise and efficient solutions. While the concept of using neural networks for solving PDEs dates back to Dissanayake & Phan-Thien (1994), the advent of the deep learning era has revitalized research interest and facilitated innovation in this domain.

Numerous studies have suggested diverse neural network architectures, diverse loss functions and even diverse activation functions for solving PDEs. For instance, the deep Galerkin method (DGM) (Sirignano & Spiliopoulos, 2018) and the physics-informed neural networks (PINNs) (Raissi et al., 2019) proposed a simple but general framework to solve PDEs using residuals of equations as their losses. To minimize these losses, the DNNs are trained via stochastic gradient descent that employs random sampling at spatial points in the domain. Boundary conditions are imposed either by an explicit integration into the networks (Berg & Nyström, 2018) or through penalization in the losses. The former approach needs reformulations of networks that may restrict their approximation performance. The latter method relaxes boundary conditions with some penalty coefficients as hyperparameters, the tuning of which remains an art and needs further systematic studies. It is noteworthy that the inclusion of high-order derivatives in the loss function can significantly compromise the solution accuracy. To mitigate this issue, Lyu et al. (2022) proposed the deep mixed residual method (MIM) in which high-order PDEs are recast as first-order PDE systems, therefore improving the approximation accuracy. Instead of using the strong forms of PDEs, the deep Ritz method (Yu et al., 2018) applies their weak formulations to convert PDEs into optimization problems, whose solutions are naturally obtained using backward propagation within deep learning framework. Recently, the convergence analysis of PINNs and Deep Ritz method has been investigated in detail (Lu et al., 2022; Duan et al., 2022; Jiao et al., 2022). Many other DNN-based PDE solvers are also available by using, for instance, minimax formulation (Zang et al., 2020), backward stochastic differential equations (Han et al., 2018), operator splitting (Lan et al., 2023). Besides, learning differential operators has also attracted increasing interests in recent years (Lu et al., 2021; Li et al., 2021). Fourier Neural Operator (FNO), regarded as a kernel operator method, offering an approach to learning mappings between infinite-dimensional spaces of functions through neural networks. Rooted in the principles of Fourier analysis, FNO can learn the resolution-invariant solution operator for PDEs. This methodology presents a promising avenue in the exploration of kernel operator methods for the effective representation and learning of complex mathematical operators, particularly within the realm of partial differential equations.

In DNN-based methods, the representations of solution operators are quite crucial, as it will affect the training efficiency and the stability of the methods. A recently proposed boundary integral network (BINet) (Lin et al., 2023b) presented a convolution representation of the solutions to elliptic PDEs using Green's functions. Specifically, given an elliptic PDE $\mathcal{L}u(x) = 0$ for

$x \in \Omega$ and $u(x) = g(x)$ on $x \in \partial\Omega$, where $\Omega$ is a bounded domain in $\mathbb{R}^d$ and the operator $\mathcal{L}$ possesses a known fundamental solution, BINet shows that the solution operator can be represented using the single and double layer potentials ($\mathcal{S}[h](x) := -\int_{\partial\Omega} G_0(x, y)h(y)ds_{\boldsymbol{y}}$ and $\mathcal{D}[h](x) := -\int_{\partial\Omega} \frac{\partial G_0(x,y)}{\partial \boldsymbol{n_y}} h(y)ds_{\boldsymbol{y}}$ with $\boldsymbol{n_y}$ denoting out normal of $\partial\Omega$ at $\boldsymbol{y}$) for some appropriate continuous function $h$ defined on $\partial\Omega$. Moreover, the density function $h$ can be approximated by a neural network that can be trained by comparing the solution with neural network involvement and the boundary condition. This method was further implemented with general Green's function that can be learned using another neural network (Lin et al., 2023a), as detailed in Appendix A. It is observed that the approximation accuracy of solution operators and the training efficiency can be improved if more mathematical prior knowledge could be built into the network architectures or loss functions.

### 1.2 SCOPE AND CONTRIBUTION OF THIS WORK

We summarize our contribution as follows:

- We proposed a new method called Neural Evolutionary Kernel Method (NEKM) for solving a class of time-dependent semi-linear PDEs. Our proposed method synergistically integrates operator splitting, boundary integral techniques, and DNNs to establish evolutionary blocks to approximate solution operators. The mathematical prior knowledge are built into each block through a convolution operation and nonlinear activations, which are adapted for the PDEs of concern. The boundary integral representation facilitates the lower regularity assumption on the solutions and serious reduction of network parameters and sampling points, leading to improved training efficiency. In addition, the proposed method can be applied to problems in complex domains as well as on manifolds.

- This approach can be combined with other time discretization schemes that possess structure preserving properties, for example, energy stability.

- We proposed a method for calculating singular boundary integrals arising from fundamental solutions, improving the training efficiency.

- We tested our method on the heat equations, Allen-Cahn equations on complex domains and manifolds, demonstrating its high accuracy and generalizability across different domains.

## 2 METHOD

In this section, we detail our method in two parts: the main framework of NEKM and a variant that possesses some properties. Let $\Omega \subset \mathbb{R}^d$ be a bounded domain, we consider this evolutionary PDE with initial condition and Dirichlet boundary condition (other conditions can also be handled by our method with a little modification of the following method):

$$\begin{cases} \frac{\partial u}{\partial t} = \tilde{\mathcal{L}}u(x, t), & x \in \Omega, \ t \geq 0 \\ u(x, 0) = f(x), & x \in \Omega, \\ u(x, t) = g(x, t), & x \in \partial\Omega, \ t > 0, \end{cases} \tag{1}$$

where $\tilde{\mathcal{L}}$ is an linear or semi-linear operator composed of differential operations (on spatial variables) and other operations acting on the value of $u$. For example, $\tilde{\mathcal{L}}u = \Delta u$ or $\tilde{\mathcal{L}}u = \Delta u + \phi(u)$, here $\phi$ is a scalar function. In fact, we need to impose some restrictions on the operator $\tilde{\mathcal{L}}$ and function $g$ :

**Assumption 1.** *(1.1) $\tilde{\mathcal{L}}$ could be written as $\tilde{\mathcal{L}} = \mathcal{L} + \phi$, where $\mathcal{L}$ is an linear operator that the fundamental solution for $\mathcal{I} - k\mathcal{L}(k \in \mathbb{R}_+)$ can be obtained and $\phi$ is a scalar function of $u$.*

    *(1.2) $g$ should be independent of time variable $t$. Otherwise, although our methods still work, we need to perform neural network training at each time step.*

### 2.1 MAIN FRAMEWORK

Denote $\tau \in \mathbb{R}$ as the time step size, then we can use $u^0(x), u^1(x), ..., u^n(x), ...$ to approximate the solution $u$ at time $t = 0, t = \tau, ..., t = n\tau, ...$ . The first equation of 1 can be discreted as

$\frac{u^{n+1}-u^n}{\tau} = \tilde{\mathcal{L}}u^{n+1}$, which can be further rewritten as $(\mathcal{I} - \tau\tilde{\mathcal{L}})u^{n+1} = u^n$, here $\mathcal{I}$ is the identity operator. We aim to determine a rule that reflects the transition from the solution at the current time step to the solution at the next time step.

Depending on whether $\phi$ is equal to 0 or not, we can classify equation 1 into the following two cases.

**Case 1: Linear Equations**  In this case, we want to solve PDE 1 with $\tilde{\mathcal{L}} = \mathcal{L} + \phi$ and $\phi = 0$, here $\mathcal{L}$ is an linear operator that the fundamental solution for $\mathcal{I} - k\mathcal{L}(k \in \mathbb{R}_+)$ can be obtained.

Suppose $u^n$ ($n \in \mathbb{Z}_{\geq 0}$) is known (since $u^0 = f$ is known and we can do induction), then $u^{n+1}$ is the sum of solutions $v_{n+1,1}$ and $v_{n+1,2}$ of these two equations:

$$\begin{cases} (\mathcal{I} - \tau\mathcal{L})v_{n+1,1} = u^n & \text{in } \Omega, \\ v_{n+1,1} = 0 & \text{on } \partial\Omega, \end{cases} \tag{2}$$

$$\begin{cases} (\mathcal{I} - \tau\mathcal{L})v_{n+1,2} = 0 & \text{in } \Omega, \\ v_{n+1,2} = g^{n+1} & \text{on } \partial\Omega, \end{cases} \tag{3}$$

where $g^n(x) := g(x, n\tau)$. Due to the linearity of operator $\mathcal{L}$, this decomposition is obvious.

To achieve high-accuracy solutions of equation 3 on arbitrary domains, we consider utilizing the BINet method by Lin et al. (2023b) to solve it and get the solution $v_{n+1,2}$. Suppose the fundamental solution $G_0$ corresponding to operator $(\mathcal{I} - \tau\mathcal{L})$ is known, then we have

$$v_{n+1,2}(x) = -\int_{\partial\Omega} G_0(x,y)\mathcal{N}_{\mathcal{S}}^{n+1}(y,\theta)ds_{\boldsymbol{y}}, \tag{4}$$

or

$$v_{n+1,2}(x) = -\int_{\partial\Omega} \frac{\partial G_0(x,y)}{\partial \boldsymbol{n_y}}\mathcal{N}_{\mathcal{D}}^{n+1}(y,\theta)ds_{\boldsymbol{y}}, \tag{5}$$

where the neural network $\mathcal{N}_{\mathcal{S}}^{n+1}(y,\theta)$ or $\mathcal{N}_{\mathcal{D}}^{n+1}(y,\theta)$ for single or double layer potential is used to approximated the density function.

And the solution $v_{n+1,1}$ of equation 2 can be gained by the inhomogeneous term $u^n$ and Green's function $G$ as this form (see Appendix B for details):

$$v_{n+1,1}(x) = \int_{\Omega} G(x,y)u^n(y)dy. \tag{6}$$

Notice that the network $\mathcal{N}_{\mathcal{S}}^{n+1}$ or $\mathcal{N}_{\mathcal{D}}^{n+1}$ in expression 4 or 5 may be different for different $n$, which means we need to train a new neural network for each step. But if we impose the previously mentioned restrictions on the function $g$, we will discover the following wonderful fact. Since the boundary conditions of equation 3 are uniform for different $n$, the neural network $\mathcal{N}_{\mathcal{S}}^{n+1}$ or $\mathcal{N}_{\mathcal{D}}^{n+1}$ are the same for different $n$. This indicates that the rules to obtain $u^{n+1}$ from $u^n$ are the same for every $n$:

$$u^{n+1}(x) = v_{n+1,1}(x) + v_{n+1,2}(x) = \int_{\Omega} G(x,y)u^n(y)dy - \int_{\partial\Omega} G_0(x,y)\mathcal{N}_{\mathcal{S}}(y,\theta)ds_{\boldsymbol{y}}, \tag{7}$$

or

$$u^{n+1}(x) = v_{n+1,1}(x) + v_{n+1,2}(x) = \int_{\Omega} G(x,y)u^n(y)dy - \int_{\partial\Omega} \frac{\partial G_0(x,y)}{\partial \boldsymbol{n_y}}\mathcal{N}_{\mathcal{D}}(y,\theta)ds_{\boldsymbol{y}}. \tag{8}$$

This also means that if the Green's function is known, we only need to train the neural network once to obtain all the $u^{n+1}$ based on these uniform recursive formulas. If the Green's function is unknown, we only need to train another neural network one more time based on the BI-GreenNet method by Lin et al. (2023a) to obtain the Green's function of operator $(\mathcal{I} - \tau\mathcal{L})$ and domain $\Omega$.

**Remark.**

- When the domain $\Omega$ is a square domain, We might be able to use some fast algorithms to solve equation 2 instead of using the Green's function.
- The details of BINet method and BI-GreenNet method can be found in Appendix A.

- The numerical computation of the integrals 4 and 5 are not trivial. In general, for well-known singularities such as those caused by functions like $\ln(x)$ and $\frac{1}{\sqrt{x}}$, we can directly integrate or apply integration by parts. For singularities caused by unfamiliar functions, we can employ an asymptotic expansion technique to transform them into well-known singular functions for further processing. Here is an example to describe the detail implementation.

To compute equation 4 for $G_0(x, x') = \frac{1}{2\pi\tau}\mathbb{K}_0(\frac{|x-x'|}{\sqrt{\tau}})$ ($\mathbb{K}_\gamma(x)$ is the modified Bessel function of the second kind of order $\gamma$), which is the fundamental solution of $\mathcal{L} = \mathcal{I} - \tau\Delta$, we note the asymptotic series of $\mathbb{K}_0$ as following (Bender et al., 1999):

$$\mathbb{K}_0(x) = -(\ln(\frac{x}{2}) + \gamma)\mathbb{I}_0(x) + \sum_{n=1}^{\infty} \frac{(\frac{x}{2})^{2n}}{(n!)^2}(1 + \frac{1}{2} + ... + \frac{1}{n}), \tag{9}$$

where $\gamma$ is the Euler–Mascheroni constant with $\gamma \approx 0.5772156649...$ and $\mathbb{I}_0$ is the modified Bessel function of the first kind of order 0 which can be regarded as:

$$\mathbb{I}_0(x) = \sum_{n=0}^{\infty} \frac{(\frac{x}{2})^{2n}}{(n!)^2}. \tag{10}$$

For integral 4, the singularity happens when $y$ nears $x$. For these $y$, we can approximate the value of $G_0(x, y)$ according to some leading terms of $\mathbb{K}_0$ by equation 9. For $d = 2$, the variable $y$ on $\partial\Omega$ and near $x$ can be parameterized by one parameter, and we just need to deal singular integral such as $\int_{-\epsilon}^{\epsilon} \ln y h(y) dy$ where $\epsilon$ is a small positive number. This can be handled as:

$$\int_{-\epsilon}^{\epsilon} \ln y h(y) dy = \int_{-\epsilon}^{\epsilon} h(y) d(y\ln y - y)$$

$$= h(y)(y\ln y - y)|_{-\epsilon}^{\epsilon} - \int_{-\epsilon}^{\epsilon} h'(y)(y\ln y - y) dy.$$

And this form has no singularity since $y\ln(y) - y \to 0$ as $y \to 0$. For $d \geq 3$, we can parameterize $y \in (x + \epsilon\mathbb{B}^{d-1}) \bigcap \partial\Omega$ by polar coordinates (with $\epsilon$ being a small positive number), where $\mathbb{B}^n$ is the $n$-dimensional unit ball.

**Case 2: Semi-Linear Equations** In this case, we want to solve PDE 1 with $\tilde{\mathcal{L}} = \mathcal{L} + \phi$ and $\phi \neq 0$, here $\mathcal{L}$ is an linear operator that the fundamental solution for $\mathcal{I} - k\mathcal{L}(k \in \mathbb{R}_+)$ can be obtained and $\phi$ is a scalar function about $u$.

We will use the idea of operator splitting to transform equation 1 to a linear PDE and an equation independent with the spatial differential, which can be regarded as an ordinary differential equation (ODE). In our method, we will use the Strang's scheme of operator splitting for equation 1. In specifically, to solve PDE 1 in $[t^n, t^{n+1}](n \in \mathbb{Z}_{\geq 0}, t^n := n\tau)$ with a initial value $u^n$ and boundary condition $g$, we can consider the three equations:

$$\frac{\partial u^*}{\partial t} = \phi(u^*) \quad \text{for } t \in [t^n, t^{n+\frac{1}{2}}], \quad \text{with } u^*(\cdot, t^n) = u^n, \tag{11}$$

$$\frac{\partial u^{**}}{\partial t} = \mathcal{L}(u^{**}) \quad \text{for } t \in [t^n, t^{n+1}], \quad \text{with } u^{**}(\cdot, t^n) = u^*(\cdot, t^{n+\frac{1}{2}}), \tag{12}$$

$$\frac{\partial u^{***}}{\partial t} = \phi(u^{***}) \quad \text{for } t \in [t^{n+\frac{1}{2}}, t^{n+1}], \quad \text{with } u^{***}(\cdot, t^{n+\frac{1}{2}}) = u^{**}(\cdot, t^{n+1}), \tag{13}$$

where $t^{n+\frac{1}{2}} := t^n + \frac{1}{2}\tau$ and $u^*, u^{**}, u^{***}$ are functions defined on $\Omega \times [t^n, t^{n+1}]$. Note that above equations are needed to impose the appropriate boundary conditions given by function $g$. In this way, we can use function $u^{***}(\cdot, t^{n+1})$ to approximate the function $u^{n+1}$, which we desired as a numerical solution to PDE 1 at time $t = (n + 1)\tau$.

Note that the equations 11 and 13 can be integrated using standard ODE solvers either analytically or numerically. This together with the numerical solution of equation 12 using the approach in case 1 implies an alternating process for approximating the semi-linear PDE 1.

The standard framework for NEKM is shown in Figure 1.

**Unique features of the NEKM**

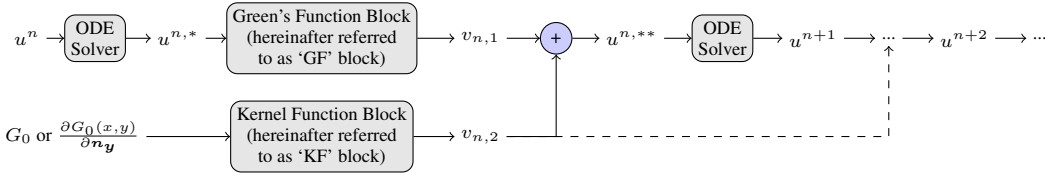

Figure 1: Schematic sketch of NEKM.

- This method effectively prescribes a network structure through boundary integral convolution by using prior knowledge from both fundamental solutions and domain geometries. Furthermore, the Green's function as a convolution kernel can be pre-trained once and reused in subsequent evolutionary process, thereby saving the training cost and improve the learning efficiency.

- Within each step, the kernel function is defined on $\partial\Omega$, which is one-dimensional lower than the whole domain. This makes the sampling and training processes more efficient.

- The boundary integral representations allow the network approximations of solutions on complex domains with low regularity assumptions. This avoids unnecessary differentiation of neural networks and facilitates their approximation accuracy. The boundary condition is naturally incorporated and no additional hyper-parameters are needed. In particular, the difficulties in the convergence of loss function and even the inability in training, which are often suffered from by other derivative based methods, may be circumvented (Appendix D.2).

## 2.2 NEKM Combined with Other Time Discretization Schemes

Due to the fact that our method only utilizes neural networks at the spatial level and is capable of solving equations in the form of $(\mathcal{I} - a\mathcal{L})u^{n+1} = f(u^n)$ ($a$ is a constant numbers, $\mathcal{L}$ is a linear operator and $f$ is a scalar function) after time discretization, NEKM can be combined with various time discretization schemes that possess desirable properties such as energy decay and energy conservation, thereby preserving the properties inherent in such schemes.

For instance, many PDEs can simulate physical phenomena that possess energy stability properties which mean the energy does not increase over time. Therefore, it is desirable for the numerical solutions of PDEs to also exhibit this important characteristic. NEKM can be combined with many energy-stable algorithms, such as convex splitting (Elliott & Stuart, 1993), invariant energy quadratization (IEQ) (Yang, 2016; Zhao et al., 2017), and scalar auxiliary variable (SAV) (Shen et al., 2019) The specific implementation details can be referred to in section 3.2.

## 3 Experiments

The heat equation is a common and important type of equation that describes the phenomenon of heat conduction. The Allen-Cahn equation is one of the basic models for the diffuse interface approach that is developed to study phase transitions and interfacial dynamic in material science (Shah & Yuan, 2011), which was originally introduced by Allen & Cahn (1979) to describe the motion of anti-phase boundaries in crystalline solids. The Allen-Cahn equation is typically expressed in one of these common forms: $u_t = \epsilon\Delta u - \frac{1}{\epsilon}F'(u)$ or $u_t = \Delta u - \frac{1}{\epsilon^2}F'(u)$, where $\epsilon$ is a small positive number and the function $F(u) = \frac{1}{4}(u^2 - 1)^2$. Hence, we choose these two kinds of PDEs as the subjects of our numerical experiments. More numerical experiments can be found in Appendix D.

In the part of neural network, the fully connected neural networks (FCNNs) or residual neural networks (ResNets) are used to approximate the density function. Specifically, we will use **ReLU** functions or **tanh** functions as the activation functions and **Adam** as the optimizer.

### 3.1 Heat Equations

In this subsection, we consider the heat equation $u_t = \Delta u$ which $u$ is the function of $x$ and $t$ with $x \in \Omega$, with different $\Omega$, initial and boundary conditions.

**Rectangular Domain** In this example, we set $\Omega = [0, \pi]^2$, $t \in [0, 0.5]$, initial condition $u(x, 0) = \sin(x_1)\sin(x_2) + 1$ and boundary condition $u(x, t) = 1, x \in \partial\Omega$ and $t \in [0, 0.5]$ which has the exact solution $u(x, t) = e^{-2t}\sin(x_1)\sin(x_2) + 1$. We use NEKM with the main framework to solve it.

Firstly, we choose $\tau = 0.1$. To get $v_{n+1,2}$ (the result of 'KF' block) using BINet method, we use the fully connected neural network with 6 hidden layers and 100 neurons per layer with **tanh** activation functions. For computing the loss function, we choose equidistant 400 sample points on $\partial\Omega$. To get $v_{n+1,1}$ (the result of 'GF' block) in this **special case**, we can use the two-dimensional fast sine transformation. Some results are shown in figure 2. To compute the error, we uniformly set up $51 * 51$ sampling points in the domain $\Omega$ and calculate the function values of the predict solution and the exact solution at time $t = 0.1$ on these points. By computing, the absolute $L^2$ error and relative $L^2$ error at $t = 0.1$ are $6.51322e - 3$ and $4.85667e - 3$, respectively. Incidentally, the absolute $L^2$ error is defined as $error_{abs} = \sqrt{\sum_{i=1}^{N}(u_{predict}(x_i) - u_{exact}(x_i))^2}/\sqrt{N}$ in this paper.

Another concern about NEKM is whether the error will increase to an unbearable level as time goes on. In this regard, we give a negative answer. To see this, we show the relative $L^2$ errors at time $t = 0.1, 0.2, 0.3, 0.4, 0.5$ in figure 3. From this figure we can find that the error grows slower and slower, and keeps within the range of high precision.

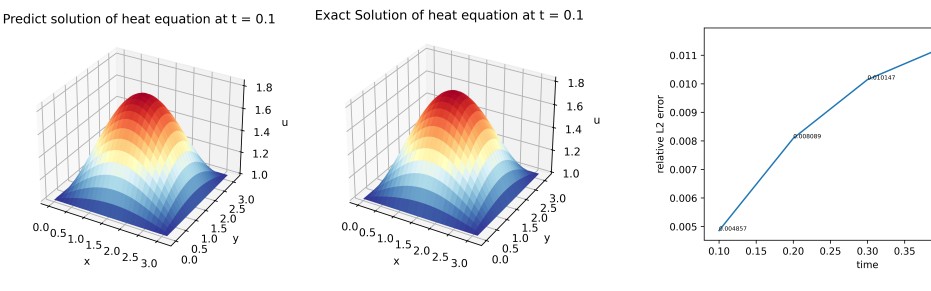

(a) Predict solution.

(b) Exact solution.

Figure 2: Solution for the heat equation at time $t = 0.1$. These figures compare the solution given by NEKM and the exact solution at time $t = 0.1$.

Figure 3: Relative errors. This figure shows the relative $L^2$ errors at different time for the predict solutions and exact solutions.

**L-shaped Domain** To illustrate the applicability of NEKM to general domains, we denote two rectangle domains by $R_1 = [-1, 0] \times [-1, 1]$ and $R_2 = [-1, 1] \times [-1, 0]$. Then $R_1 \bigcap R_2$ defines a L-shaped domain, which is the $\Omega$ in this example. And we set the initial value as $u(x, 0) = \sin(x_1)\sin(x_2)$ and the boundary condition as zero.

In this example, we choose $\tau = 0.01$. $v_{n+1,2}$ (the result of 'KF' block) can be obtained in exactly the same way as in the rectangular domain example. To get Green's function $G$ by BI-GreenNet, we use the fully connected neural network with 6 hidden layers and 150 neurons per layer with **tanh** activation functions. After gaining $G$, $v_{n+1,1}$ (the result of 'GF' block) can be obtained by equation 6. Some results are shown in figure 4.

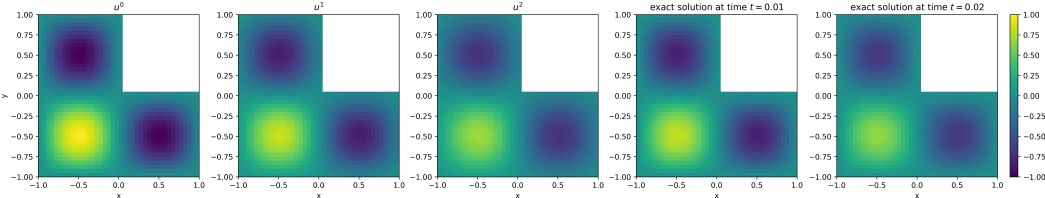

Figure 4: Solution for the heat equation at time $t = 0, 0.01, 0.02$. These figures compare the solutions given by NEKM and the exact solutions.

### 3.2 ALLEN-CAHN EQUATIONS

We aim to obtain a predictive solution for energy decay in Allen-Cahn equation $u_t = \Delta u + \frac{1}{\epsilon^2}(u - u^3), \epsilon = 0.1, x \in \Omega = [-\frac{1}{2}, \frac{1}{2}]^2, t > 0$ with initial $u(x,0) = \tanh(\frac{|x|-0.3}{0.1\sqrt{2}})$ and boundary condition $u = 1, \forall x \in \partial\Omega$ and $t > 0$. Convex splitting scheme implies that $\frac{u^{n+1}-u^n}{\tau} = \Delta u^{n+1} - su^{n+1} + su^n - \frac{1}{\epsilon^2} f'(u^n)$, where $f(u) = \frac{1}{4}(u^2-1)^2$. In order to ensure stability, we require $s - \frac{1}{\epsilon^2} f''(u) \geq 0, \forall u \in [-1,1]$, which implies $s \geq \frac{2}{\epsilon^2}$. Thus we can obtain

$$(id - \frac{\tau}{1+s\tau}\Delta)u^{n+1} = (id - \frac{\tau}{\epsilon^2(1+s\tau)}f')u^n \tag{14}$$

which is a form that can be solved by NEKM with case 1, requiring replacing $\tau$ with $\frac{\tau}{1+s\tau}$ in equation 2 and 3 and modifications of the right hand side of the first equation in equation 2.

In NEKM, we choose $\tau = 0.005$. And we need to get $v_{n+1,2}$ (the result of 'KF' block) by BINet method. For this, we use the fully connected neural network with 6 hidden layers and 150 neurons per layer with tanh activation functions. For computing the loss function, we choose equidistant 3200 sample points on $\partial\Omega$. To get $v_{n+1,1}$ (the result of 'GF' block) in this special case, we can use the 2D fast sine transformation. Then we can get the solution $u^{n+1}$ by $v_{n+1,2}$ and $v_{n+1,1}$.

To compare the results obtained by NEKM and traditional finite difference method for solving equation 14, and to validate that the NEKM framework combined with convex splitting indeed possesses the property of energy stability, we also employ finite difference methods with small $\Delta x = \Delta y = 0.005$ and the same $\tau = 0.005$ with our method to solve equation 14. If we consider the solution obtained by finite difference method with a very fine mesh as the exact solution, the errors of our method are shown in the following table 1. Figure 5 illustrates the energy evolution for both methods.

Table 1: $L^2$ Errors given by NEKM combined with convex splitting

|  | $L^2$ absolute errors | $L^2$ relative errors |
|---|---|---|
| $t = 0.005$ | $1.961500\mathrm{E}-03$ | $2.679015\mathrm{E}-03$ |
| $t = 0.010$ | $2.637368\mathrm{E}-03$ | $3.575214\mathrm{E}-03$ |
| $t = 0.015$ | $3.043784\mathrm{E}-03$ | $4.097140\mathrm{E}-03$ |
| $t = 0.020$ | $3.355984\mathrm{E}-03$ | $4.482861\mathrm{E}-03$ |
| $t = 0.025$ | $3.626770\mathrm{E}-03$ | $4.803405\mathrm{E}-03$ |
| $t = 0.030$ | $3.875158\mathrm{E}-03$ | $5.084055\mathrm{E}-03$ |
| $t = 0.035$ | $4.109637\mathrm{E}-03$ | $5.335856\mathrm{E}-03$ |
| $t = 0.040$ | $4.335455\mathrm{E}-03$ | $5.565334\mathrm{E}-03$ |
| $t = 0.045$ | $4.557162\mathrm{E}-03$ | $5.777788\mathrm{E}-03$ |
| $t = 0.050$ | $4.779471\mathrm{E}-03$ | $5.978301\mathrm{E}-03$ |

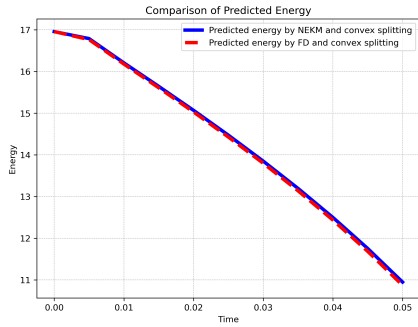

Figure 5: Value of energy functional. This figure shows the value of energy functional of two solutions.

From the above graphs, it can be observed that NEKM combined with convex splitting exhibits good accuracy and also maintains energy stability properties.

### 3.3 NEKM FOR PDEs ON SURFACES

One noteworthy fact is that our method can be extended to PDEs on surfaces, as long as the operators on the surface and their corresponding fundamental solutions exist. An example of solving the heat equation on a hemisphere is provided below.

**Heat equation** Suppose $\Omega = \{(\rho, \theta, \varphi) \in \mathbb{R}_+ \times [0, \frac{\pi}{2}] \times [0, 2\pi] : \rho = 1\}$ is the episphere with the spherical coordinate: $x = \rho \sin(\theta)\cos(\varphi), y = \rho\sin(\theta)\sin(\varphi), z = \rho\cos(\theta)$. In spherical coordinates, the Laplace-Beltrami operator instead of the common Laplacian is given by $\Delta_s = \frac{1}{\rho^2}\frac{1}{\sin^2\theta}\frac{\partial^2}{\partial\varphi^2} + \frac{1}{\rho^2}\frac{1}{\sin\theta}\frac{\partial}{\partial\theta}(\sin\theta\frac{\partial}{\partial\theta})$. In this example, we consider heat equation on episphere with initial condition $u(\theta, \varphi, t = 0) = \cos\theta$ and zero boundary condition ($\partial\Omega = \{(\rho, \theta, \varphi) \in \{1\} \times \{\frac{\pi}{2}\} \times [0, 2\pi]\}$), whose exact solution is $u(\theta, \varphi, t) = e^{-2t}\cos\theta$. NEKM with the main framework

was used to solve it. We choose $\tau = 0.1$ as the time step. Note that the detail expression of fundamental solution $G_0$ of $\mathcal{I} - k\Delta_s (k > 0)$ can be found in Appendix F and we can employ the method of images to compute the Green's function $G$ on the upper hemisphere with Dirichlet boundary conditions as follows:

$$G((\theta, \varphi), (\theta', \varphi')) = G_0((\theta, \varphi), (\theta', \varphi')) - G_0((\theta, \varphi), (\pi - \theta', \varphi')). \tag{15}$$

To get $v_{n+1,2}$ (the result of 'KF' block) using BINet method, we use the fully connected neural network with 6 hidden layers and 100 neurons per layer with **tanh** activation functions. For computing the loss function, we choose equidistant 400 sample points on $\partial\Omega$. To get $v_{n+1,1}$ (the result of 'GF' block), we can use the known Green's function $G$ and integral 6. Some results are shown in figure 6. To compute the error, we uniformly set up $80 * 320$ sampling points in the domain $\Omega$ at time $t = 0.1, 0.2, ..., 0.5$ and calculate the function values of the predict solution and the exact solution. By computing, the absolute $L^2$ error over $(\theta, \phi, t) \in [0, \frac{\pi}{2}] \times [0, 2\pi] \times [0, 0.5]$ is 0.0188670.

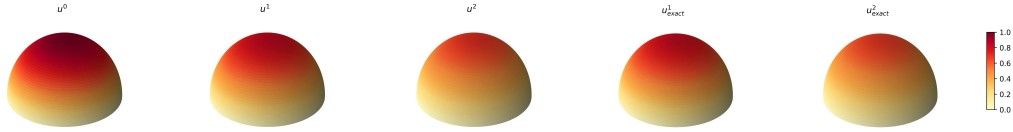

Figure 6: Solution for the heat equation at time $t = 0, 0.1, 0.2$. These figures compare the solutions given by NEKM and the exact solutions.

**Allen-Cahn equation** In this part, we employ NEKM to solve the Allen-Cahn equation $u_t(\theta, \varphi, t) = \Delta_s u(\theta, \varphi, t) + \frac{1}{\epsilon^2}(u - u^3)(\epsilon = 0.1)$ defined on episphere with zero Neumann boundary condition and random initial value between $-0.1$ and $0.1$, which simulates the processes of phase separation. We present the results of solving this equation using NEKM directly in figure 7, as the process of solving it is similar with solving the heat equation in the previous section except the ODE solver processes which are detailed in Appendix D.1. It can be observed that the solution of this equation actually simulates phase separation process.

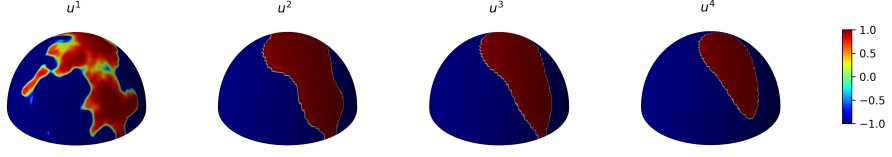

Figure 7: Predict solution for the Allen-Cahn equation. These figures show the predict solution of Allen-Cahn equation given by NEKM at time $t = 0.1, 0.2, 0.3$ and $0.4$.

## 4 CONCLUSION AND DISCUSSION

In summary, this paper presents the Neural Evolutionary Kernel Method (NEKM) for solving a class of semi-linear time-dependent PDEs. NEKM combines operator splitting, boundary integral techniques, and DNNs to establish evolutionary blocks and accurate numerical solutions in complex domains. The method can be combined with other time discretization schemes that possess desirable properties such as energy stability, thereby preserving the structure of the PDEs. Experiments on heat equations, Allen-Cahn equations in different domains confirms the high accuracy and generalizability of the method.

Extensions of the proposed method are also available, for example, solving systems of PDEs and even equations with discrete operators. These promote neural network modeling and solutions of general evolutionary problems in real world, as many realistic dynamics are represented using manifold operators or graph Laplacian. We will also consider higher-order accuracy approximations of solution operators with general boundary conditions.

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

# A  BINet method and BI-GreenNet method

In this part, we will present two novel methods that offer important contributions to our approaches in solving time-dependent PDEs. The first method is designed for solving PDEs on general domains, while the second one is specifically developed for finding Green's functions. These two methods serve as essential building blocks and provide valuable insights into the development of our approach.

## A.1  Boundary Integral Network

We introduce a method of solving PDEs combining boundary integral equations and neural networks by Lin et al. (2023b) briefly. Let $\Omega \subset \mathbb{R}^d$ be a bounded domain, we consider the PDE:

$$\mathcal{L}u(x) = 0, \tag{16}$$

$\mathcal{L}$ can be arbitrary operator, as long as its fundamental solution (in $\mathbb{R}^d$) can be obtained (in fact, the fundamental solution only depends on the operator and the fundamental solutions of most common operators are already known). The PDE is defined in $\Omega$ or $\Omega^C := \mathbb{R}^d \backslash \bar{\Omega}$ and we consider the Dirichlet type of boundary condition $u|_{\partial\Omega} = g$. Other types of boundary conditions can be easily handled in BINet with a small modification of the loss function.

The following theorem (Kellogg, 1953) in potential theory is the basis of BINet method.

**Theorem 1.** *For any continuous function $h$ defined on $\partial\Omega$, the single layer potential is defined as*

$$\mathcal{S}[h](x) := -\int_{\partial\Omega} G_0(x, y) h(y) ds_{\boldsymbol{y}}, \tag{17}$$

*and the double layer potential is defined as*

$$\mathcal{D}[h](x) := -\int_{\partial\Omega} \frac{\partial G_0(x, y)}{\partial \boldsymbol{n_y}} h(y) ds_{\boldsymbol{y}}, \tag{18}$$

*with $\boldsymbol{n_y}$ denotes out normal of $\partial\Omega$ at $\boldsymbol{y}$, $G_0(x, y)$ is the fundamental solution of equation 16. Then, both single layer potential and double layer potential satisfy equation 16. And for all $x_0 \in \partial\Omega$, we have*

$$\lim_{x \to x_0} \mathcal{S}[h](x) = \mathcal{S}[h](x_0), \tag{19}$$

$$\lim_{x \to x_0 \pm} \mathcal{D}[h](x) = \mathcal{D}[h](x_0) \mp \frac{1}{2} h(x_0). \tag{20}$$

*where $x \to x_0^-$ and $x \to x_0^+$ mean converging in $\Omega$ and $\Omega^C$ respectively.*

From theorem 1, we can find that $\mathcal{S}[h]$ or $\mathcal{D}[h]$ satisfies PDE 16 for all continuous function $h$ defined on $\partial\Omega$, so we just need to find the most appropriate '$h$' to let the single or double potential satisfies the boundary condition.

Thus, we can use a neural network $\mathcal{N}(y, \theta)$ to estimate function $h$ with the learning parameter $\theta$, and use the following loss function for training:

$$L(\theta) = \begin{cases} ||\mathcal{S}[\mathcal{N}(:, \theta)](x) - g(x)||^2_{\partial\Omega}, & \text{single layer potential} \\ ||(\frac{1}{2}\mathcal{I} + \mathcal{D})[\mathcal{N}(:, \theta)](x) - g(x)||^2_{\partial\Omega}, & \text{double layer potential(Interior)} \\ ||(-\frac{1}{2}\mathcal{I} + \mathcal{D})[\mathcal{N}(:, \theta)](x) - g(x)||^2_{\partial\Omega}, & \text{double layer potential(Exterior)} \end{cases} \tag{21}$$

where $\mathcal{I}$ is the identity operator, 'Interior' means the PDE is defined on a bounded domain while 'Exterior' means the PDE is defined on the complement of a bounded domain.

The BINet method offers several advantages. Firstly, it leverages the mathematical characteristics of the equation's solution. Secondly, it can address higher-dimensional problems and handle both internal ($\Omega$) and external ($\Omega^C$) regional issues, while preserving the unsmoothness of the solution on the boundary. Thirdly, it avoids errors and instability resulting from high-order differentiation of neural networks. Fourthly, it requires fewer sampling points and is easy to train, as only sample points on the boundary are needed. Finally, it can be used to solve for the Green function, which is challenging to obtain for general domains.

### A.2 Boundary Integral-Green Network

The Green's function is a crucial tool in the analysis and solution of PDEs. However, obtaining an analytical solution of the Green's function for a general operator and domain is nearly impossible due to its dependence on both the operator and the shape of the domain. Furthermore, the variables involved in the Green's function are often high-dimensional, making it difficult for traditional numerical methods to provide effective solutions. Despite these challenges, the development of methods for obtaining the Green's function for various operators and domains has remained an active research area, as it can provide valuable insights into the behavior of solutions to PDEs and facilitate the development of numerical methods for solving these equations. In this subsection, we propose a method, boundary integral-Green network by Lin et al. (2023a) for computing the Green's function based on the BINet approach.

Let $\Omega \subset \mathbb{R}^d$ be a bounded domain, we consider the PDE:

$$\begin{cases} \mathcal{L}u(x) = f(x) \text{ in } \Omega, \\ u(x) = g(x) \text{ on } \partial\Omega, \end{cases} \tag{22}$$

where $\mathcal{L}$ is a differential operator. To find the Green's function $G(x, y)$ of this equation, we firstly define $H(x, y) = G(x, y) - G_0(x, y)$, where $G_0$ is the fundamental solution of the operator $\mathcal{L}$ (in space $\mathbb{R}^d$). Note that the definition of Green's function and its difference from the fundamental solution can be found in Appendix B of this paper. Consider that $G(x, y)$ vanishes when $x$ goes to $\partial\Omega$, we only care about the value of $H(x, y)$ for $(x, y) \in \Omega \backslash \partial\Omega \times \Omega$. Then we can get the function for $H$ as following:

$$\begin{cases} \mathcal{L}_y H(x, y) = 0, & \forall x \in \Omega \backslash \partial\Omega, y \in \Omega, \\ H(x, y) = -G_0(x, y), & \forall x \in \Omega \backslash \partial\Omega, y \in \partial\Omega. \end{cases} \tag{23}$$

One observation is that for each $x \in \Omega \backslash \partial\Omega$, $H(x, \cdot)$ is a function with one variable in $\Omega \subset \mathbb{R}^d$, then the BINet method can be using to find it. Therefore we can use the form of single or double potential to approximate the solution $H$. That is,

$$H(x, y) \approx \mathcal{S}[h](x, y) := - \int_{\partial\Omega} G_0(y, z) h(x, z) ds_{\boldsymbol{z}}, \tag{24}$$

or

$$H(x, y) \approx \mathcal{D}[h](x, y) := - \int_{\partial\Omega} \frac{\partial G_0(y, z)}{\partial \boldsymbol{n_z}} h(x, z) ds_{\boldsymbol{z}} \tag{25}$$

can be used to approximate the solution of equation 23. In this case, if we use a neural network $\mathcal{N}(x, y; \theta)$ to estimate function $h(x, y)(x \in \Omega \backslash \partial\Omega, y \in \partial\Omega)$, the loss function can be set as

$$L(\theta) = \begin{cases} \sum_{i=1}^{N} |\mathcal{S}[\mathcal{N}(:, \theta)](x_i, y_i) + G_0(x_i, y_i)|^2, & \text{single layer potential} \\ \sum_{i=1}^{N} |\mathcal{D}[\mathcal{N}(:, \theta)](x_i, y_i) + \frac{1}{2}\mathcal{N}(x_i, y_i; \theta) + G_0(x_i, y_i)|^2, & \text{double layer potential} \end{cases} \tag{26}$$

where $\{x_i, y_i\}_{j=1}^{N}$ are $N$ points randomly sampled in $\Omega \backslash \partial\Omega \times \partial\Omega$. After finding the most 'appropriate' function $h$ through training, we can get the solution $H$ and therefore the Green's function $G$ by $G = H + G_0$.

We are now able to find the Green's function of an operator in a general domain as long as the fundamental solution of the operator is known. This is not only an excellent achievement, but also provides some algorithmic support for our later work on solving time-evolving PDEs.

## B Fundamental Solution and Green's Function

In this part, we make some supplementary introductions to the fundamental solution and Green's function. Let $\Omega \subset \mathbb{R}^d$ be a bounded domain, $\Omega^C := \mathbb{R}^d \backslash \bar{\Omega}$ and $\Omega^* := \Omega$ or $\Omega^C$. $\mathcal{L}$ is a differential operator.

**Definition 1.** *A function $\delta(x)$ is called a d-dimensional $\delta$-function if $\delta(x) \simeq \begin{cases} 0, & x \neq \vec{0}, \\ \infty, & x = \vec{0}, \end{cases}$ and which is also constrained to satisfy the identity $\int_{\mathbb{R}^d} \delta(x)dx = 1$. For all function $f$ that is continuous at $a \in \mathbb{R}^d$ we have $\int_{\mathbb{R}^d} f(x)\delta(x - a)dx = f(a)$.*

**Definition 2.** *A function $G_0(x, y)$ is called the fundamental solution corresponding to equations $\mathcal{L}u(x) = 0$ if $G_0(x, y)$ is symmetric about $x$ and $y$ and $G_0(x, y)$ satisfy $\mathcal{L}_y G_0(x, y) = \delta(x - y)$, where $(x, y) \in \mathbb{R}^d \times \mathbb{R}^d$ and $\mathcal{L}_y$ is the differential operator L which acts on component y.*

**Definition 3.** *A function $G(x, y)$ is called the Green's function corresponding to problem*

$$\begin{cases} \mathcal{L}u(x) = f(x) & in\ \Omega^*, \\ u(x) = g(x) & on\ \partial\Omega^*, \end{cases}$$

*if $G(x, y)$ is a 2d-dimensional function satisfying*

$$\begin{cases} \mathcal{L}_y G(x, y) = \delta(x - y), & \forall x, y \in \Omega^*, \\ G(x, y) = 0, & \forall x \in \Omega^*, y \in \partial\Omega^*. \end{cases}$$

*In addition, if the type of boundary condition in the problem is changed, the boundary conditions that Green's function needed to satisfy must also be changed to the corresponding zero boundary condition.*

By the definition of fundamental solution and Green's function, the differences between them is clear and easy to understand. Fundamental solution is only depend on the operator while Green's function is depend both on the operator and the boundary condition.

## C   OPERATOR SPLITTING

Some knowledge about the operator splitting of ODEs will be given here. And it can be extended to operator splitting of PDEs.

We focus our attention on the case of two linear operators. Let us consider the Cauchy problem :

$$\frac{\partial U(t)}{\partial t} = AU(t) + BU(t),\ t \in [0, T], U(0) = U_0,$$

whereby, the initial function $U_0$ is given and A and B are supposed to be linear operators in the Banach-space $X$ with $A$ and $B : X \to X$. Splitting methods assume that the mathematical problem can be split into two or more terms (Omer et al., 2017). We denote by $U(t) = e^{(A+B)t}U_0$ is the solution (Yazici, 2010) at the time $t$ of the differential equation. One issue in computing the solution $U(t)$ is that it may not be feasible or extremely challenging to accurately evaluate the exponential mapping. Consequently, numerical methods are frequently used to approximate the flow map EXP.

Firstly, we describe the first order operator splitting method, which is called Lie-Trotter splitting. Lie-Trotter splitting is introduced as a method, which solves two subproblems sequentially on subintervals $[t^n, t^{n+1}]$, where $n = 0, 1, ..., N - 1$, $t^0 = 0$ and $t^N = T$. Assume that $t^{n+1} - t^n = \tau$ for all $n$.The Lie-Trotter's scheme is that $U^{n+1} = e^{A\tau}e^{B\tau}U^n$ or $U^{n+1} = e^{B\tau}e^{A\tau}U^n$, where $n = 0, 1, ..., N - 1, U^0 := U_0$. And the splitting error is computed by

$$\begin{aligned} SE_{Lie-Trotter} &= \mathcal{I} + \tau(A + B) + \frac{1}{2}\tau^2(A + B)^2 \\ &\quad - (\mathcal{I} + \tau B + \frac{1}{2}\tau^2 B^2)(\mathcal{I} + \tau A + \frac{1}{2}\tau^2 A^2) \\ &\quad + O(\tau^3) \\ &= \frac{1}{2}\tau^2(AB - BA) + O(\tau^3) \\ &= O(\tau^2), \end{aligned}$$

where $\mathcal{I}$ is the identical operator. This implies that the Lie-Trotter splitting is a second order approach for the local truncation error, then we call it as the first order splitting method for the global error.

The Strang's scheme is regarded as a second order splitting. Take the same notation as before, the Strang's scheme is that $U^{n+1} = e^{B\frac{\tau}{2}}e^{A\tau}e^{B\frac{\tau}{2}}U^n$ or $U^{n+1} = e^{A\frac{\tau}{2}}e^{B\tau}e^{A\frac{\tau}{2}}U^n$. And the splitting

error is computed by

$$
\begin{aligned}
SE_{Strang} &= \mathcal{I} + \tau(A+B) + \frac{1}{2}\tau^2(A+B)^2 \\
&\quad - (\mathcal{I} + \frac{\tau}{2}B + \frac{1}{8}\tau^2 B^2)(\mathcal{I} + \tau A + \frac{1}{2}\tau^2 A^2)(\mathcal{I} + \frac{\tau}{2}B + \frac{1}{8}\tau^2 B^2) \\
&\quad + O(\tau^3) \\
&= O(\tau^3),
\end{aligned}
$$

where $\mathcal{I}$ is the identical operator. This implies that the Strang splitting is a third order approach for the local truncation error, then we call it as the second order splitting method for the global error.

# D    ADDITIONAL NUMERICAL EXPERIMENTS

The Allen-Cahn equation is a type of nonlinear equation that is often used to test the accuracy and stability of numerical methods. In other words, for a traditional numerical method, we hope that it has the property of stability. However, our NEKM method does not have such concerns because it does not discretize in space.

## D.1    ALLEN-CAHN EQUATION IN THE RECTANGLE DOMAIN

In this subsection, we consider an Allen-Cahn equation in the rectangle domain as following:

$$
\begin{cases}
u_t = 0.1\Delta u + 10(u - u^3), & x \in \Omega = [-\frac{1}{2}, \frac{1}{2}]^2 \text{and } t \in [0, 0.1] \\
u(x,0) = \tanh(\frac{|x|-0.3}{0.1\sqrt{2}}) =: f, & x \in \Omega \\
u(x,t) = 1 =: g, & x \in \partial\Omega \text{ and } t > 0
\end{cases}
\tag{27}
$$

This equation has the form of equation 1 with $\tilde{\mathcal{L}} = 0.1\Delta + \phi$ and we can decompose it as $\tilde{\mathcal{L}} = \mathcal{L} + \phi$ with $\mathcal{L} = 0.1\Delta$ and $\phi = 10(u - u^3)$. Now, we can use NEKM of case 2 to solve it. Meanwhile, we use PINNs to solve this equation and compared the solutions obtained by these two DNN-based methods with the exact solution to obtain the magnitude of the errors. It is worth mentioning that we used the forward Euler method with a very dense mesh and time and spatial step sizes that satisfied the stability condition to solve this equation, and the solution obtained from this is taken as the exact solution.

In NEKM, we choose $\tau = 0.01$. To solve equation 12, we use NEKM of case 1. And we need to get $v_{n+1,2}$ (the result of 'KF' block) by BINet method. For this, we use the fully connected neural network with 6 hidden layers and 150 neurons per layer with **tanh** activation functions. For computing the loss function, we choose equidistant 3200 sample points on $\partial\Omega$. To get $v_{n+1,1}$ (the result of 'GF' block) in this special case, we can use the two-dimensional fast Fourier transformation. Then we can solve equation 12 by NEKM with case 1. Remaining work is to find the solution of equation 11 and equation 13. If we can do this, the solution of this Allen-Cahn equation can be obtained. These two equations can be reduced to the following form:

$$
\frac{dy}{dt} = \frac{1}{\epsilon}(y - y^3), t \in [t^n, t^{n+1}], y(t^n) = y^n,
\tag{28}
$$

which we want to find the value of function $y$ at $t = t^{n+1}$. We can achieve our goal in following ways. From equation 28 we can find that

$$
\int_{y^n}^{y^{n+1}} \frac{dy}{y - y^3} = \int_{t^n}^{t^{n+1}} \frac{dt}{\epsilon},
\tag{29}
$$

where $y^{n+1} := y(t^{n+1})$. Note that $\frac{1}{y-y^3} = \frac{1}{y} - \frac{1}{2(y+1)} - \frac{1}{2(y-1)}$ is unintegrable on an interval which contains $-1, 0$ or $1$, we can conclude that $y^n$ and $y^{n+1}$ must be both located in one of the four intervals: $(-\infty, -1), (-1, 0), (0, 1), (1, \infty)$. Equation 29 implies that

$$
\ln\left(\frac{|y^2|}{|y^2 - 1|}\right)\Big|_{y^n}^{y^{n+1}} = \frac{2(t^{n+1} - t^n)}{\epsilon}.
\tag{30}
$$

Thus we can find the value of $y^{n+1}$ by equation 30 that

$$y^{n+1} = \frac{y^n}{\sqrt{(y^n)^2 + (1-(y^n)^2)e^{-\frac{2(t^{n+1}-t^n)}{\epsilon}}}}. \tag{31}$$

Now we obtain the exact solution of equation 28, therefore the equation 11 and 13 can be solved. Thus we can find the numerical solution of equation 27 by NEKM since equation 11-13 are solvable. Some results are shown in figure 8.

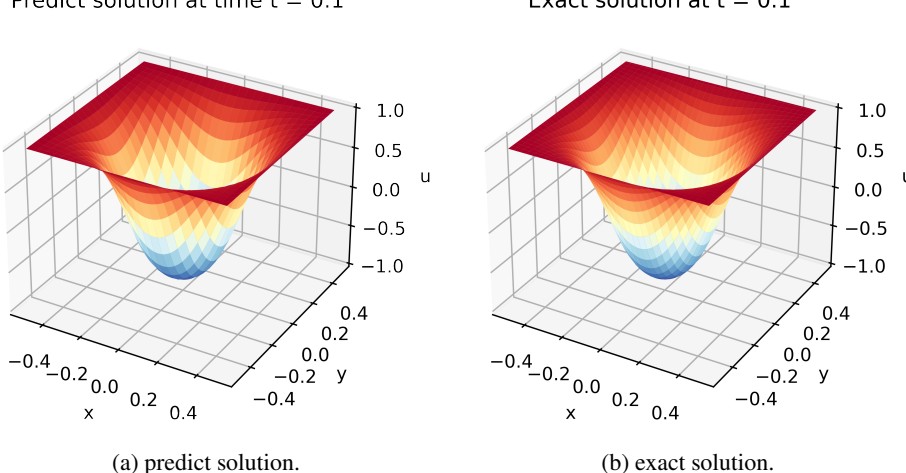

Predict solution at time t = 0.1  Exact solution at t = 0.1

(a) predict solution.      (b) exact solution.

Figure 8: Solution for the Allen-Cahn equation at time $t = 0.1$. These figures compare the solution given by NEKM and the exact solution at time $t = 0.1$.

Also, we solve the Allen-Cahn equation using PINNs. We use the fully connected neural network with 6 hidden layers and 100 neurons per layer with **tanh** activation functions. For computing the loss function, we choose equidistant 400 sample points for the initial condition, equidistant 1000 sample points for the boundary condition and random 1800 samples for the PDE.

To compute the error, we uniformly set up $51 * 51$ sampling points in domain $[-\frac{1}{2}, \frac{1}{2}]^2$ and calculate the function values of the predict solution by NEKM and the exact solution on these points at time $t = 0.01, 0.02, 0.03, 0.04, 0.05$, respectively. The result is given by table 2 as following.

Table 2: $L^2$ Errors given by NEKM and PINNs

|  | NEKM (absolute) | PINNs (absolute) | NEKM (relative) | PINNs (relative) |
|---|---|---|---|---|
| $t = 0.01$ | $3.8034E-3$ | $1.2634E-2$ | $5.2215E-3$ | $1.7345E-2$ |
| $t = 0.02$ | $5.3697E-3$ | $1.2648E-2$ | $7.3395E-3$ | $1.7289E-2$ |
| $t = 0.03$ | $6.3527E-3$ | $1.1874E-2$ | $8.6544E-3$ | $1.6176E-2$ |
| $t = 0.04$ | $7.1423E-3$ | $1.0653E-2$ | $9.5653E-3$ | $1.4470E-2$ |
| $t = 0.05$ | $7.5646E-3$ | $9.1909E-3$ | $1.0245E-2$ | $1.2448E-2$ |

And we compute the errors of numerical solutions on $\Omega \times [0, 0.1]$. The absolute $L^2$ errors of NEKM and PINNs are 0.00717583 and 0.00870301, respectively. And the relative $L^2$ errors of NEKM and PINNs are 0.00973051 and 0.0118014, respectively. The results show that for this example, our method is slightly better than PINNs method. Moreover, our method performs significantly better than PINNs in the interval near the initial time. In fact, our method can solve PDEs with high accuracy in more general domain, while PINNs cannot achieve this (we will provide an example later).

In fact, the above example can be regarded as simulating the evolution of a circular interface with certain rules given by the initial conditions to some extent (in fact, when $\epsilon$ tends to 0, the solution

of the Allen-Cahn equation tends to the characteristic function whose interface evolves by motion by mean curvature flow (Fischer et al., 2020)). Below, we modify the initial conditions in order to simulate the evolution of a petal-shaped interface.

Consider this Allen-Cahn equation:

$$\begin{cases} u_t = 0.1\Delta u + 10(u - u^3), & x \in \Omega = [-\frac{1}{2}, \frac{1}{2}]^2 \text{and } t \in [0, 0.1] \\ u(x, 0) = \tanh(\frac{4|x| - 0.3\cos(6\arctan(\frac{y}{x})) - 1.2}{0.1\sqrt{2}}), & x \in \Omega \\ u(x, t) = 1, & x \in \partial\Omega \text{ and } t > 0 \end{cases} \quad (32)$$

We select $\tau = 0.01$ and solve the problem exactly as we did in the previous example, and some of the results are shown below (figure 9, figure 10). It can be seen from the results that the petal-shaped interface gradually evolves into a circular interface, which is consistent with the fact that the mean curvature flow changes.

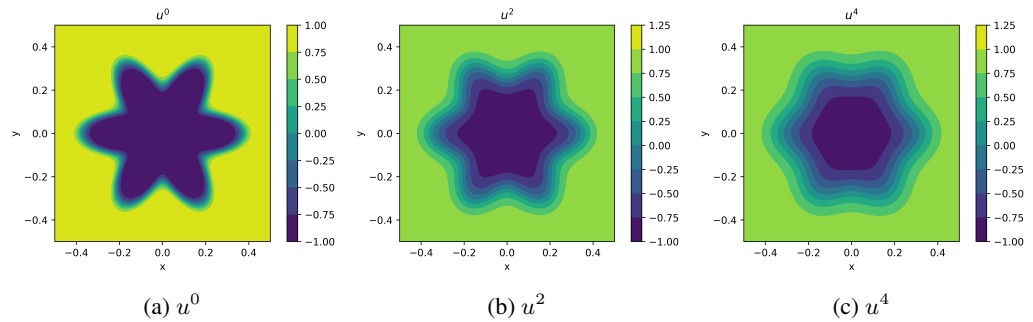

(a) $u^0$            (b) $u^2$            (c) $u^4$

Figure 9: Predict solution for the Allen-Cahn equation. These figures show the predict solution of Allen-Cahn equation given by NEKM at time $t = 0, t = 0.02, t = 0.04$

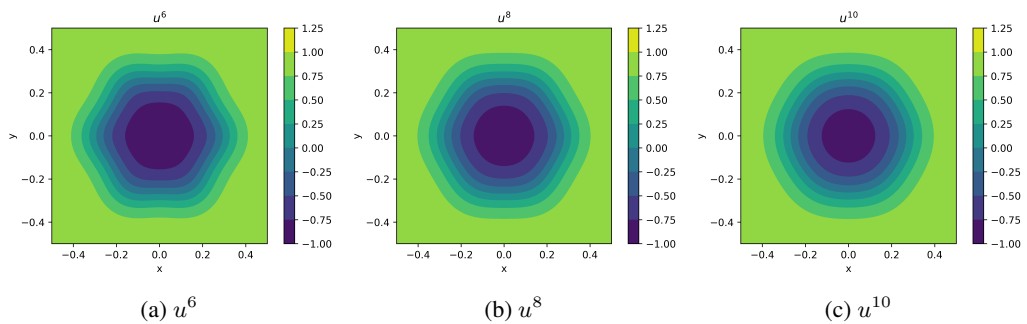

(a) $u^6$            (b) $u^8$            (c) $u^{10}$

Figure 10: Predict solution for the Allen-Cahn equation. These figures show the predict solution of Allen-Cahn equation given by NEKM at time $t = 0.06, t = 0.08, t = 0.1$

### D.2 ALLEN-CAHN EQUATION IN THE DISK DOMAIN

In this subsection, we solve the Allen-Cahn equation on a disk domain using NEKM method and show that PINNs did not perform well in this example, while our method still had good performance. We consider an Allen-Cahn equation in a disk domain as following:

$$\begin{cases} u_t = 0.5\Delta u + 2(u - u^3), & x \in \Omega = 0.2\mathcal{B}^2 \text{and } t \in [0, 0.1] \\ u(x, 0) = \tanh(100\frac{|x| - 1.5\cos(6\arctan(\frac{y}{x})) - 6}{0.5\sqrt{2}}), & x \in \Omega \\ u(x, t) = 1, & x \in \partial\Omega \text{ and } t > 0 \end{cases} \quad (33)$$

where $\mathcal{B}^2$ is the 2-dimensional ball in $\mathbb{R}^2$.

In NEKM, we choose $\tau = 0.1$. To solve equation 12, we use NEKM of case 1. And we need to get $v_{n+1,2}$ (the result of 'KF' block) by BINet method. For this, we use the fully connected

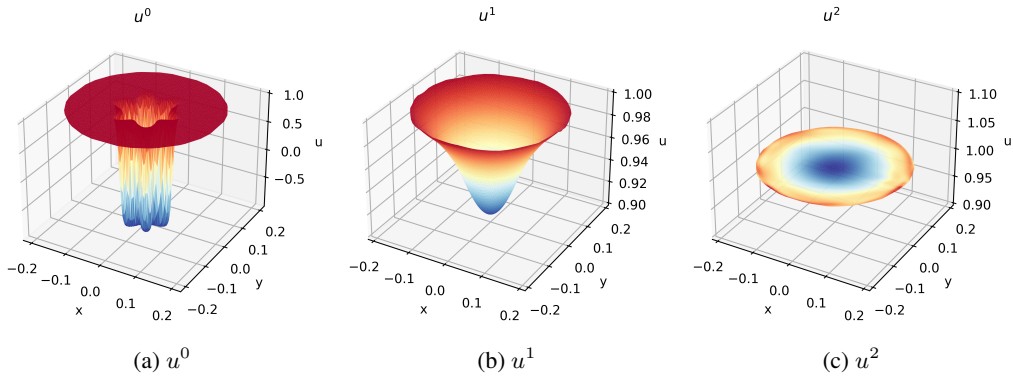

(a) $u^0$            (b) $u^1$            (c) $u^2$

Figure 11: Predict solution for the Allen-Cahn equation. These figures show the predict solution of Allen-Cahn equation given by NEKM at time $t = 0, t = 0.1, t = 0.2$

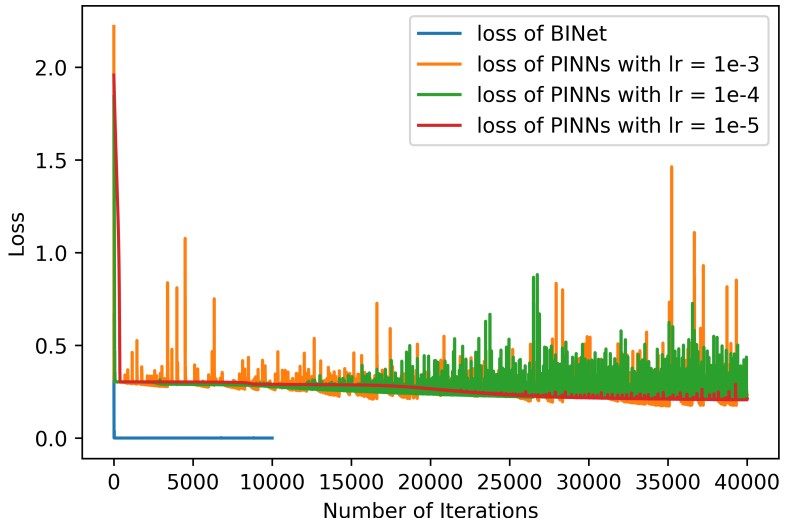

Figure 12: Value of loss functions. This figure shows the value of loss function with number of iterations under different methods or learning rate of Adam optimizer.

neural network with 6 hidden layers and 150 neurons per layer with **tanh** activation functions. For computing the loss function, we choose equidistant 1600 sample points on $\partial\Omega$. To get Green's function $G$ by BI-GreenNet, we use the fully connected neural network with 6 hidden layers and 150 neurons per layer with **tanh** activation functions. For computing the loss function, we choose equidistant 200 sample points $\{y_i\}_{j=1}^{200}$ on $\partial\Omega$. For every 500 epochs, 100 new $x$ are randomly generated. Then $\{x_i\}_{i=1}^{100}$ and $\{y_i\}_{i=1}^{200}$ form a set of sample points $\{(x_i, y_i)\}_{i=1}^{20000}$. After gaining $G$, $v_{n+1,1}$ (the result of 'GF' block) can be obtained by equation 6. And equation 11 and 13 can be solved analytically (with details in Appendix D). Thus we can obtain the numerical solution by NEKM. Some of the results are shown below (figure 11).

We find the fact that PINNs did not perform well in this example. Specifically, we use the fully connected neural network with 6 hidden layers and 100 neurons per layer with **tanh** activation functions (in fact, we tried other sizes of networks, and it didn't get any better). For computing the loss function, we choose equidistant nearly 300 sample points for the initial condition, equidistant 4000 sample points for the boundary condition and random 20000 samples for the PDE (in fact, we tried other sizes of sample sets, and it didn't get any better). But we find that the loss function can not converge to zero or a small positive number, as shown in Figure 12.

Therefore the solution given by PINNs can not be considered a good numerical solution for equation 33. Some possible reasons for this are that in the disk domain, geometric characteristics such as curvature and normal vector may affect the solution of the equation, and PINNs method can not capture these information well. At the same time, other optimizers may bring better solutions to PINNs method. However, our method is not subject to these limitations, and the loss function can also converge well for the domain of general shape.

## E A REVIEW OF PINNs

As we have compared our method with PINNs several times in Appendix D, we briefly introduce the general framework of PINNs in this section. Consider the PDE

$$\begin{cases} \frac{\partial u}{\partial t} = \mathcal{N}[u], & (x,t) \in \Omega \times [0,T], \\ u(x,0) = f(x), & x \in \Omega, \\ u(x,t) = g(x,t), & (x,t) \in \partial\Omega \times [0,T], \end{cases}$$

where $\mathcal{N}$ is an operator combined with differential and other operations and $T$ is a positive number. The main idea of the PINNs method by Raissi et al. (2019) is to use a neural network $u(x,t;\theta)$ to approximate the solution $u$, where $\theta$ represents the trainable parameters in the neural network. Then we can use the automatic differentiation tool to calculate the derivative of $u(x,t;\theta)$ and get the value of $\frac{\partial u}{\partial t}$ and $\mathcal{N}[u]$ at arbitrary $(x,t)$. Thus we can define the loss function as $L(\theta) = ||\frac{\partial u(\cdot;\theta)}{\partial t} - \mathcal{N}[u(\cdot;\theta)]||^2_{\Omega \times [0,T]} + \beta_1 ||u(\cdot;\theta) - f(\cdot)||^2_{\Omega \times \{0\}} + \beta_2 ||u(\cdot;\theta) - g(\cdot;\theta)||^2_{\partial\Omega \times [0,T]}$. Here $\beta_1$ and $\beta_2$ are two hyper-parameters. By minimizing the loss function $L$, PINNs will get the approximation solution of the PDE. In addition, the computation of these $L^2$ norms is accomplished through random sampling over relevant domains.

## F FUNDAMENTAL SOLUTION OF SOME OPERATORS

The fundamental solution for operator $\mathcal{I} - \tau\Delta$ is

$$G_0(x,x') = \frac{1}{2\pi\tau}\mathbb{K}_0(\frac{|x-x'|}{\sqrt{\tau}}), \tag{34}$$

where $\mathbb{K}_\gamma(x)$ is the modified Bessel function of the second kind of order $\gamma$. Note that the fundamental solution can be obtained by the fundamental solution of screened Poisson equation easily.

The fundamental solution for operator $\Delta_s - k\mathcal{I}(k > 0)$ with $\Delta_s$ be the Laplacian on sphere is (Tanios et al., 2019)

$$G_{0,k}((\theta,\varphi),(\theta',\varphi')) = -\frac{1}{4\pi}\sum_{l=0}^{\infty}\frac{2l+1}{l(l+1)+k}P_l(\cos(\gamma)), \tag{35}$$

where $\cos(\gamma) = \cos\theta\cos\theta' + \sin\theta\sin\theta'\cos(\varphi-\varphi')$ and $P_l(x)$ is the Legendre polynomial of degree $l$. To estimate the value of this fundamental solution, we can choose $l' \in \mathbb{R}_+$ such that $l'(l'+1) >> k$ and approximate $G_{0,k}$ according to (Tanios et al., 2019):

$$\begin{aligned} G_{0,k}((\theta,\varphi),(\theta',\varphi')) \approx &- \frac{1}{4\pi k} - \frac{1}{4\pi}\sum_{l=1}^{l'-1}[\frac{2l+1}{l(l+1)+k} - \frac{2l+1}{l(l+1)}]P_l(\cos(\gamma)) \\ &+ \frac{1}{4\pi}\log(\frac{e}{2}(1-\cos(\gamma))). \end{aligned} \tag{36}$$

Thus the fundamental solution which we desired for operator $\mathcal{I} - \tau\Delta_s$ is $G_0 = -\frac{1}{\tau}G_{0,\frac{1}{\tau}}$. This fundamental solution is essentially the Green's function over the entire spherical space.

