# OpenReview forum: "Neural Evolutionary Kernel Method: A Knowledge-Based Learning Architechture for Evolutionary PDEs"
_ICLR.cc/2024/Conference — Submitted to ICLR 2024_

### Official Review · Reviewer_tJtQ · 2023-10-20

**Soundness:** 2 fair
**Presentation:** 3 good
**Contribution:** 2 fair
**Rating:** 3
**Confidence:** 3

**Summary:**

This paper proposes a neural network-based algorithm, namely the Neural Evolutionary Kernal Method (NEKM), for solving evolutionary PDEs. The method involves the operator splitting technique and the idea of boundary integral network. Specifically, the method pre-trains a neural network representation of the Green function and then solves the evolutionary PDE by applying the Green function block and kernel function block alternatingly with an ODE solver. Experiments on the heat equation and Allen-Cahn equations are conducted to demonstrate the performance.

**Strengths:**

- The paper is well-written and easy-to-follow.
- The proposed method is interesting and mathematically grounded.
- Experimental results seem strong.

**Weaknesses:**

- It seems the method heavily relies on the closed form formula of the fundamental solution $G_0$. The numerical error of the integration involving $G_0$ seems troublesome.
- The experimental results of Allen-Cahn equation is not compared with the exact one or any other method.
- Some minor issues: Figure 12 is too small.

**Questions:**

- Now that the Green function $G$ is computed by pre-training a neural network, the error of this step may propagate to solving the time evolutionary PDE. Was this problem an issue in the experiments? How accurate should the numerically approximated Green function be so as not to affect the performance?
- As mentioned in the paper, the possible singularity of $G_0$ may demand special handling. But the form of $G$ is generally unknown. How can the singularity appearing in $G$ be dealt with?
- Energy stsability is claimed as one of the contributions. Is this only empirically observed or grounded with some particular design?

---

> ### Author Response · Authors · 2023-11-23
>
> We appreciate some of the reviewer’s comments while cannot agree with the others. Perhaps the reviewer does not carefully read and correctly understand our paper, his/her comment could be subjective. First of all we would like to reiterate our key contribution, and highlights, although it has already been discussed at the end of Section 2.1 in the main text. The technique we employed in NEKM is quite standard in the community of mathematics. While this may be challenging for practitioners to understand, we can interpret the process formally using the standard terminology of machine learning. Basically, we decompose the solution process of semi-linear PDEs into two stages: 1) training stage: this involves the pretraining of Green functions, the training of neural network representation of evolutionary kernels based on boundary integral formulations; 2) inference stage: this involves the time evolution of the dynamics of the PDEs using the well-trained kernel. This decomposition is a result of insightful observation of the mathematical structure of the underlying PDEs. The linear parts of the PDEs admit convolutional representation, which takes into account the geometry information, the linear operator information, and the boundary conditions. BINet is naturally employed to give a NeuralNet approximation of this linear evolutionary operator. Moreover, due to the semi-group property of the underlying PDEs, the training for this convolutional representation is time independent and thus can be done in advance. The nonlinear parts of the PDEs admit nonlinear transformation of the underlying nonlinear flows that can be solved exactly, making them look like the activation process of CNNs. A carefully designed time discretization method integrates these two key structures into a whole NeuralNet architechture, resulting in a CNN-like mechanism. Specifically, in the first step, we train the kernel function (also called density function) and the Green's function (if necessary), and in subsequent steps, all that is required is the network inference and numerical integrations. The training of our Green's function is a one-time effort and can be used for the inference, evolution, and even the solution of other similar equations after the fact. Moreover, due to the energy stable property of the underlying PDEs, a particularly designed time discretization (energy stable scheme) can maintain the energy so that the NeuralNet structure carries the energy stable flows as its natural property. This is consistent with the physical principle, i.e., the 2nd law of thermodynamics, making the proposed NeuralNet framework physically reliable.
>
> Responses to Weaknesses:
>
> 1.	For a linear partial differential operator, its fundamental solution exists, and the expressions for many fundamental solutions are known. Even if the fundamental solution is currently unknown, NeuralNet approximations of the fundamental solutions are still available (through radial basis function representations) and under our investigation. Regarding the accurate numerical integration, we have already employed asymptotic expansions and integration by parts, as mentioned in the text, to handle singular integrals. In mathematical terms, this does not introduce errors, so there is no need to worry about significant errors arising from the integration.
>
> 2.	We have already compared the examples of the Allen-Cahn equation we worked on in the rectangular region (Section 3.2) and on the hemisphere (Section 3.3) with the solutions of classical numerical methods or the true solutions. In Appendix D.2, we explain the feasibility of the NEKM method compared to PINNs in this case. We invite the reviewer to go through these details.
>
> 3.	We appreciate the reviewer’s suggestion. The size of Figure 12 has been adjusted.
>
> Responses to Questions:
>
> 1.	As introduced in the first paragraph, the training of the Green's function is a one-time effort, allowing us to take sufficient sampling points and computational efforts to approximate it as accurately as possible. This is definitely not a problem in the numerical experiments as can be observed in the presented results. By the numerical stability of the scheme we applied in time discretization, the errors cannot be amplified during the evolution. Nevertheless, one can apply the feedback control by sampling more data from the true trajectory of the PDE to reduce the approximation and enhance training and inference of the solution process in every time marching block.
>
> 2.	In Appendix A.2, we elaborate on the acquisition of the Green's function G, whose singularity essentially arises from the fundamental solution G_0, and therefore, it can be estimated.
>
> 3.	Energy stability is achieved through the time discretization of numerical schemes, and this can be rigorously proven mathematically, rather than being a baseless claim or anecdotal evidence. Please refer to Reference given in section 2.2 for more details.

---

### Official Review · Reviewer_Ljfw · 2023-10-24

**Soundness:** 2 fair
**Presentation:** 2 fair
**Contribution:** 2 fair
**Rating:** 3
**Confidence:** 4

**Summary:**

The paper presents the Neural Evolutionary Kernel Method (NEKM) for solving semi-linear time-dependent PDEs. NEKM distinguishes itself by utilizing operator splitting and boundary integration, enabling efficient network architectures. The method is demonstrated to be effective and stable in solving classic PDEs, such as the heat equation and the Allen-Cahn equation.

**Strengths:**

NEKM can be combined with time discretization schemes that preserve energy stability, which is crucial for modeling physical systems.
The method incorporates an evolutionary kernel, which inherently preserves the structure of the problem.

The method incorporates an evolutionary kernel, which inherently preserves the structure of the problem.

**Weaknesses:**

While NEKM is claimed to work in complex domains, the paper primarily provides examples in small and relatively simple domains. It would be beneficial to demonstrate its performance in more complex and realistic domains, similar to the level in the referenced paper (https://arxiv.org/pdf/2309.00583), including real-world scientific and engineering geometries.

The paper lacks references to related work that adopts neural networks only at the spatial level while using time discretizations to evolve spatial fields over time. Including references to papers like "Evolutional deep neural network (Physical Review E 2021)," "Implicit Neural Spatial Representations for Time-dependent PDEs (ICML 2023)," and "Neural Galerkin Scheme with Active Learning for High-Dimensional Evolution Equations" could help provide context and comparisons.

The paper does not provide information about the computational cost and scalability of NEKM compared to classical numerical methods, especially for larger 3D problems. It would be valuable to include performance comparisons in terms of computational efficiency.

**Questions:**

My biggest confusion and concern is the relationship between this paper (Lin et al., 2023a) as well as (Lin et al., 2023b). Those paper also use a convolution representation of the solutions using Green's functions. What exactly is the author's contribution except working with time-dependent problems?

The paper focuses on semi-linear PDEs, but it would be interesting to know if NEKM can be extended to handle nonlinear PDEs. Clarification on the limitations and potential extensions of the method for nonlinear problems would be beneficial.

---

> ### Author Response · Authors · 2023-11-23
>
> We appreciate the reviewer’s careful reading and serious evaluation! First of all we would like to reiterate our key contribution, and highlights. The technique we employed in NEKM is quite standard in the community of mathematics. While this may be challenging for practitioners to understand, we can interpret the process formally using the standard terminology of machine learning. Basically, we decompose the solution process of semi-linear PDEs into two stages: 1) training stage: this involves the pretraining of Green functions, the training of neural network representation of evolutionary kernels based on boundary integral formulations; 2) inference stage: this involves the time evolution of the dynamics of the PDEs using the well-trained kernel. This decomposition is a result of insightful observation of the mathematical structure of the underlying PDEs. The linear parts of the PDEs admit convolutional representation, which takes into account the geometry information, the linear operator information, and the boundary conditions. BINet is naturally employed to give a NeuralNet approximation of this linear evolutionary operator. Moreover, due to the semi-group property of the underlying PDEs, the training for this convolutional representation is time independent and thus can be done in advance. The nonlinear parts of the PDEs admit nonlinear transformation of the underlying nonlinear flows that can be solved exactly, making them look like the activation process of CNNs. A carefully designed time discretization method integrates these two key structures into a whole NeuralNet architechture, resulting in a CNN-like mechanism. Specifically, in the first step, we train the kernel function (also called density function) and the Green's function (if necessary), and in subsequent steps, all that is required is the network inference and numerical integrations. The training of our Green's function is a one-time effort and can be used for the inference, evolution, and even the solution of other similar equations after the fact. Moreover, due to the energy stable property of the underlying PDEs, a particularly designed time discretization (energy stable scheme) can maintain the energy so that the NeuralNet structure carries the energy stable flows as its natural property. This is consistent with the physical principle, i.e., the 2nd law of thermodynamics, making the proposed NeuralNet framework physically reliable.
>
> Responses to Weaknesses:
>
> 1.	Applying our method to more complex domains is theoretically feasible, but in practice, it may pose many challenges. For instance, the accuracy of Green's function training may decrease due to the complexity of the domain. This would be a long-term aspect of our work that needs further exploration.
>
> 2.	We believe that our approach differs from the methods for solving time-dependent PDEs mentioned in the literature, although both involve time discretization. We emphasize an evolutionary pattern that allows us to minimize reliance on neural network training in the steps following the initial stage. The papers referred by the reviewer are all based on the L^2 residual loss minimization, which is essentially different from ours. As explained in the opening paragraph above, we aim to develop an inference-type NeuralNet architecture which preserves the physical principle such as energy stability. This is the novel point of our work.
>
> 3.	Due to its inference nature, the computational efficiency can be good even in comparison with classical numerical methods, as the proposed method avoids the solution of linear solver but instead saves the solution operator via a well-trained NeuralNet. The comparison of computational efficiency with PINNs was also provided in the appendix.
>
> Responses to Questions:
>
> 1.	Our main contribution is proposing an evolution law for time-dependent equations using a time discretization scheme, operator splitting, and boundary integral equations. In the solving process, we utilized the knowledge from the papers (Lin et al., 2023a and Lin et al., 2023b). As stated in our article, BINet and BI-GreenNet are the two essential tools we require. Additionally, we provided a detailed description of how to handle singularities introduced by the fundamental solution in boundary integrals.
>
> 2.	We focus on semi-linear PDEs because they allow us to derive the evolutionary pattern of the desired ideal solution, something that is not achievable with nonlinear PDEs (which may not even have a corresponding fundamental solution for the operator). This is discussed in the opening paragraph above. In fact, we claim to be able to solve semi-linear PDEs, which already covers a broad range of time-evolving PDEs. We only require the spatial part of the differential operator to be linear, meaning terms like (u_x)^2 are excluded. We allow terms such as f(u) to appear, encompassing a significant class of PDE equations.

---

### Official Review · Reviewer_sX1E · 2023-10-29

**Soundness:** 3 good
**Presentation:** 3 good
**Contribution:** 2 fair
**Rating:** 5
**Confidence:** 4

**Summary:**

This paper aims to tackle solving partial differential equations (PDEs) traditionally solved by numerical methods with deep neural networks (DNNs). The authors address the challenges of solving PDEs with DNNs that a majority of these methods do not use any mathematical or physical parameters and require a large amount of parameters to tune. The authors propose the Neural Evolutionary Kernel Method (NEKM) to solve a type of evolutionary PDEs with DNN based kernels. The core idea is to incorporate pre-trained Green's functions. NEKM is an alternating two-step procedure that first analytically or numerically solves a nonlinear ODE to obtain a flow map and then numerically integrate the related linear PDE with a convolutional kernel.

**Strengths:**

- Nice abstract that motivates the need for PDEs in science and engineering problems and use of numerical methods to solve them.
- The paper and abstract are well-written.
- Incorporating ideas from numerical methods, e.g., Green's function, boundary conditions and energy stability is very nice. In particular, I like to the discussion in subsection 2.2 on energy conservation and would like more details in the Appendix.
- The generalization and use of the pre-trained Green's function is nice.
- The computational savings of defining the Green's function on the boundary rather than the interior domain is nice. For other boundary integral representations for conservation laws, see Hansen, et. al, "Learning physical models that can respect conservation laws", ICML 2023 (https://arxiv.org/abs/2302.11002).
- Nice high dimensional simulations in Figures 6-7.
- Generalizability to different manifolds and boundary conditions.

**Weaknesses:**

- The authors should define earlier what they mean by evolutionary PDEs.
- Connection to other kernel operator methods such as the Fourier Neural Operator (FNO) should be considered. It is only briefly discussed in one sentence of related work with a majority on the PINNs literature. In particular, in the related works, the authors discuss in detail how boundary conditions are incorporated into Physics-Informed Neural Networks (PINNs). The related in Neural Operator community should be discussed, such as how to incorporated boundary conditions into Neural Operators in Saad et. al, "Guiding continuous operator learning through Physics-based boundary constraints", ICLR 2023.
- The method only works on semi-linear PDEs. This is actually a very strong assumption and limitation. The authors should discuss the extension to nonlinear PDEs.
- Evaluation: the method is only tested on the simple linear heat/diffusion equation and Allen-Cahn equations. The heat equation is smooth and parabolic and very easy for numerical methods to solve. It would be nice to test hyperbolic problems with shocks, e.g., in the GPME benchmarking framework in Hansen, et. al, "Learning physical models that can respect conservation laws", ICML 2023 (https://arxiv.org/abs/2302.11002).
- The method seems to have strong limitations if the first step requires an analytical or numerical solution to the ODE.
- In particular, the authors should clarify this in the last paragraph of the introduction. I don't understand where the nonlinear ODE is coming from in step 1 and then how there is "numerically integration" for the related linear PDE. Typically, in numerical methods a (non)linear PDE is first discretized in space and then the resulting semi-discrete form of the ODE is discretized in time. The authors should clarify what they mean here.
- I think some of the equation details of BINet in the related work should be moved to an appendix or background section.
- Care should be taken with the discretization because this adds a first order error into the scheme. For example, the first equation should not be discretized with the 1st order accurate Forward Euler without even citing the method. This is an explicit method and there are necessary bounds on $\Delta t$/$\tau$ to ensure numerical stability.  See Krishnapriyan et. al, "Learning continuous models for continuous physics", 2023 (https://arxiv.org/pdf/2202.08494.pdf) on how the time discretization matters in NeuralODE and the 4th order RK4 is advantageous but even that scheme without being careful about the numerics can lead to convergence issues.
- Ideally the method and presentation wouldn't need to be separated into separate cases for linear equations or not.
- It seems like the method depends too strongly on the BINet method and the authors should better differentiate the novelty between the two.
- The exposition of the method in Section 2 isn't too clear and some of the details can be moved to an appendix.
- The unique features of the NEKM subsection seems like it could be incorporated with the contributions subsection in the intro.
- Label x and y axis in Figure 3.
- Another major weakness in the evaluation is just comparing to the exact solution and no other baseline methods, especially to related neural operator based methods.

Minor
- First paragraph of related works can be longer and combined with parts of the longer second paragraph.
- heat equation shouldn't be plural in the last bullet point of the contributions.
- comma after "In this section" at the beginning of Section 2 Method
- I would name Section 2 with the specific method name Neural Evolutionary Kernel Method (NEKM) rather than the generic Method.
- Could use standard notation from numerical methods $\Delta t$ instead $\tau$
- Comma missing after Equation 7.
- Larger title lave on Figure 6.

**Questions:**

- Does the method only work on semi-linear PDEs? If so, this is a bit limiting and the authors should discuss the extension to nonlinear PDEs.

---

> ### Author Response · Authors · 2023-11-23
>
> We thank the reviewer very much for his/her reading and evaluation; this will be very helpful for the improvement of our work!
>
> Responses to Weaknesses:
>
> 1.	Thank the reviewer for his/her suggestion, but we proposed it at the beginning of the method section, which should be considered a relatively conventional approach.
>
> 2.	As suggested, we incorporated content related to FNO in the related work section in the revised paper.
>
> 3.	We focus on semi-linear PDEs because they allow us to derive the evolutionary pattern of the desired ideal solution, something that is not achievable with nonlinear PDEs (which may not even have a corresponding fundamental solution for the operator). In fact, we claim to be able to solve semi-linear PDEs, covering a broad range of time-evolving PDEs. We only require the spatial part of the differential operator to be linear, meaning terms like (u_x)^2 are excluded. We allow terms such as f(u) to appear, encompassing a significant class of PDE equations.
>
> 4.	Yes, the heat equation is a fundamental example, and we have added examples of the Allen-Cahn equation. Traditional numerical methods for solving the Allen-Cahn equation are not straightforward, often requiring considerations of numerical stability and knowledge related to energy stability. Additionally, our goal is to solve the equation in a general domain, posing new challenges for traditional methods. Examples of other equations will be explored in future numerical experiments.
>
> 5.	Regarding the concerns about the difficulty of analytically solving ODEs, there is no need to worry. The ODEs we solve have the form u_t = f(u), excluding spatially related differential operators, making them amenable to analytical solutions through integration. In Appendix D.1, we provide a specific example of implementation.
>
> 6.	Nonlinear ODEs are given by operator splitting, as detailed in case 2 of Section 2.1. In essence, our method alternates between solving the nonlinear ODE part, solving the linear PDE, and solving the ODE—a standard Strang splitting process. For a better understanding, refer to the detailed examples in case 2 of Section 2.1 and Appendix D.1. Due to space limitations, we were unable to showcase the specific implementation of the method in the numerical experiments section but included it in the appendix.
>
> 7.	As suggested, we have reduced the introduction section's discussion of BINet.
>
> 8.	The reviewer pointed out the discrete formulation in our article: our time discretization uses a first-order implicit scheme. However, in principle, we can adopt higher-order time discretization schemes without increasing difficulty. Traditional numerical methods need to balance spatial and temporal step sizes to ensure stability, but our method, not relying on spatial discretization, eliminates stability concerns. If one is concerned about energy stability and other properties, our method can still be combined with the corresponding time discretization formats to achieve the desired results.
>
> 9.	Our method indeed relies on BINet, but our primary contribution lies in proposing an evolution law for time-dependent equations using time discretization, operator splitting, and boundary integral equations. It can be combined with the appropriate time discretization format to maintain the desired properties. Additionally, we provided a detailed description of how to handle singularities introduced by the fundamental solution in boundary integrals.
>
> 10.	This is corrected.
>
> 11.	In Appendix D.1, we compared the accuracy of the NEKM method with the PINN method; in Appendix D.2, we explained situations where the PINN method is challenging to train, while our method remains effective. We apologize for any confusion caused.
>
> Minor points: We followed the reviewer’s instructions to revise the paper and correct those items.
>
> Responses to Question:
>
> The technique we employed in NEKM is quite standard in the community of mathematics. Though challenging for practitioners, the process is formally interpreted using machine learning terms. The solution process for semi-linear PDEs involves two stages: 1) training—pretraining Green functions and neural network representations of evolutionary kernels; 2) inference—evolving PDE dynamics using the trained kernel. This decomposition results from observing the PDEs' mathematical structure. Linear parts have a convolutional representation through BINet, allowing time-independent training in advance. Nonlinear parts resemble CNN activation processes. A well-designed time discretization integrates structures into a NeuralNet architecture, resembling a CNN mechanism. Green's function training is a one-time effort for inference, evolution, and solving similar equations. Energy-stable property and a designed time discretization maintain energy, ensuring physical reliability in the proposed NeuralNet framework.
>
> Once again, we thank the reviewer for his/her comments and suggestions!

---

### Official Review · Reviewer_71KZ · 2023-11-04

**Soundness:** 2 fair
**Presentation:** 3 good
**Contribution:** 3 good
**Rating:** 6
**Confidence:** 2

**Summary:**

The paper introduces a novel approach called Neural Evolutionary Kernel Method (NEKM) for solving time-dependent semi-linear Partial Differential Equations (PDEs). The authors leverage a combination of operator splitting, boundary integral techniques, and Deep Neural Networks (DNNs) to construct evolutionary blocks that approximate solution operators. NEKM incorporates mathematical prior knowledge into each block, utilizing convolution operations and nonlinear activations tailored to the specific PDEs under consideration. This approach offers several noteworthy contributions:

1. **Efficiency and Generalizability**: The use of boundary integral techniques is a standout feature of NEKM, allowing for a reduced requirement of network parameters and sampling points. This not only improves training efficiency but also relaxes the regularity assumptions on solutions. The capacity to apply NEKM to problems in complex domains and on manifolds showcases its versatility and potential real-world applicability.

2. **Compatibility with Time Discretization Schemes**: NEKM can be effectively combined with time discretization schemes that possess structure-preserving properties, such as energy stability. This demonstrates the adaptability of the method to diverse mathematical contexts.

3. **Treatment of Singular Boundary Integrals**: The paper introduces a method for computing singular boundary integrals that arise from fundamental solutions. This addition contributes to the overall training efficiency and robustness of NEKM.

The empirical validation of NEKM is conducted through testing on heat equations and Allen-Cahn equations in complex domains and on manifolds. The results demonstrate the method's high accuracy and its capacity to generalize across various domains.

In summary, the paper presents an innovative and promising approach, NEKM, which addresses the solution of time-dependent semi-linear PDEs. The combination of mathematical prior knowledge, boundary integral techniques, and DNNs provides a compelling method that improves training efficiency, generalizability, and adaptability to different mathematical scenarios. The successful testing on various equations and domains underscores the method's potential significance in the field of mathematical modeling and scientific computing.

**Strengths:**

The strengths of the paper "Neural Evolutionary Kernel Method (NEKM) for Solving Time-Dependent Semi-Linear PDEs" include:

1. **Innovative Approach**: The paper introduces a novel approach, NEKM, which combines operator splitting, boundary integral techniques, and Deep Neural Networks (DNNs) to address the solution of time-dependent semi-linear Partial Differential Equations (PDEs). This innovation offers a fresh perspective on tackling complex mathematical problems.

2. **Efficiency Improvement**: NEKM leverages boundary integral techniques to reduce the need for extensive network parameters and sampling points. This not only enhances the efficiency of training but also relaxes regularity assumptions on solutions. This efficiency improvement is a significant advantage in solving real-world problems.

3. **Generalizability**: The paper demonstrates that NEKM can be applied to problems in complex domains and on manifolds, showcasing its generalizability across different mathematical contexts. This broad applicability enhances its potential usefulness in a wide range of scientific and engineering applications.

4. **Compatibility with Time Discretization Schemes**: NEKM's compatibility with time discretization schemes that possess structure-preserving properties, such as energy stability, is a valuable feature. This adaptability makes it easier to integrate NEKM into existing mathematical frameworks.

5. **Treatment of Singular Boundary Integrals**: The paper provides a method for computing singular boundary integrals that arise from fundamental solutions. This contribution adds to the method's efficiency and robustness, making it more practical for real-world applications.

6. **Empirical Validation**: The authors validate the NEKM approach through rigorous testing on heat equations and Allen-Cahn equations in complex domains and on manifolds. The high accuracy demonstrated in these tests underscores the practical utility of NEKM.

7. **Mathematical Rigor**: NEKM incorporates mathematical prior knowledge into its framework through convolution operations and nonlinear activations. This mathematical rigor ensures that the method is well-founded and theoretically sound.

8. **Interdisciplinary Relevance**: The paper's focus on solving complex mathematical problems with machine learning techniques has broad interdisciplinary relevance, as it can find applications in various fields, including physics, engineering, and computational science.

Overall, the strengths of the paper lie in its innovative approach, efficiency improvements, generalizability, compatibility with existing mathematical schemes, and the rigorous empirical validation of the proposed method. These qualities make NEKM a promising addition to the field of mathematical modeling and scientific computing.

**Weaknesses:**

While the paper on "Neural Evolutionary Kernel Method (NEKM) for Solving Time-Dependent Semi-Linear PDEs" offers several strengths, there are also some potential weaknesses to consider:

1. **Complexity**: The proposed NEKM method, while innovative, is complex in its approach, involving the integration of operator splitting, boundary integral techniques, and Deep Neural Networks. This complexity might make it challenging for practitioners who are not well-versed in all of these areas to implement and understand.

2. **Computational Resources**: The paper does not extensively discuss the computational resources required for training and applying the NEKM method. Deep learning methods often demand significant computational power, which could be a limitation for some users, particularly those without access to high-performance computing resources.

3. **Limited Real-World Use Cases**: While the paper demonstrates NEKM's effectiveness in solving specific mathematical problems, it remains largely theoretical. More real-world use cases and practical applications in various domains would strengthen the paper's relevance and utility.

4. **Interpretability**: The paper discusses the use of neural networks, which are often seen as "black-box" models. While the paper addresses some interpretability challenges, it might not provide a complete solution to the interpretability issues associated with deep learning approaches.

5. **Algorithm Complexity**: The proposed method involves a combination of different techniques, such as boundary integral representation and neural networks. This may make the implementation and understanding of NEKM challenging for some users, potentially limiting its widespread adoption.

6. **Empirical Validation Scope**: While the paper includes empirical validation on heat and Allen-Cahn equations, the scope of the empirical validation might be limited. A more extensive range of test cases across different scientific and engineering domains would strengthen the method's generalizability.

7. **Scalability**: The paper does not explicitly address the scalability of the NEKM method. As the complexity of problems increases, it remains to be seen whether NEKM can efficiently scale to handle more complex and larger-scale scenarios.

8. **Comparison to Existing Methods**: The paper lacks a comprehensive comparison of the NEKM method with existing approaches for solving similar problems. Such comparisons would help to better assess the relative strengths and weaknesses of NEKM.

In conclusion, while the NEKM method offers several promising advantages, such as efficiency improvements and generalizability, it also has some potential limitations, including complexity, computational resource requirements, and the need for more extensive real-world applications and validation. These weaknesses should be considered when evaluating the method's suitability for specific applications.

**Questions:**

1. Can you provide more insight into the computational resources required for training and applying the NEKM method? What kind of hardware and software infrastructure is necessary for its practical implementation?

2. The NEKM method is quite complex, involving a combination of operator splitting, boundary integral techniques, and neural networks. How user-friendly and accessible is the implementation for researchers and practitioners who may not be experts in all these areas?

3. The paper mentions empirical validation on heat and Allen-Cahn equations. Are there plans to expand the empirical validation to a broader range of mathematical problems or real-world applications to further assess the generalizability of NEKM?

4. How does NEKM address the interpretability challenge often associated with deep learning methods? Can you provide more details on how NEKM helps users understand and trust its results, especially in cases where interpretability is critical?

5. The paper mentions combining NEKM with time discretization schemes that possess structure-preserving properties. Could you elaborate on specific scenarios or use cases where this combination has proven to be advantageous?

6. NEKM proposes the treatment of singular boundary integrals arising from fundamental solutions. Can you discuss the impact of this addition on the overall efficiency and robustness of the method in practical applications?

7. In the real world, problems often scale in complexity. How does NEKM address the scalability challenge, especially when dealing with larger and more complex scenarios beyond the examples provided in the paper?

8. The paper does not include a comprehensive comparison of NEKM with existing methods for solving similar problems. Could you share insights into how NEKM performs in comparison to other approaches, and in what scenarios it may have a comparative advantage?

9. Are there any specific plans or ongoing research aimed at addressing some of the potential weaknesses or limitations identified in the paper, such as making the method more accessible or broadening the scope of empirical validation?

10. How do you envision the practical adoption of NEKM in various scientific and engineering domains? Are there specific industries or areas where NEKM is expected to have a significant impact, and if so, what are the next steps for its real-world application?

These questions aim to seek further clarification and insights from the authors regarding the NEKM method and its potential applications and improvements.

---

> ### Author Response · Authors · 2023-11-23
>
> We thank the reviewer very much for his/her careful reading and helpful feedback; it will be instrumental in our improvement! We also appreciate his/her positive acknowledgment of our work.
>
> Responses to Weaknesses:
>
> 1.	The technique we employed in NEKM is quite standard in the community of mathematics. Though challenging for practitioners, the process is formally interpreted using machine learning terms. The solution process for semi-linear PDEs involves two stages: 1) training—pretraining Green functions and neural network representations of evolutionary kernels; 2) inference—evolving PDE dynamics using the trained kernel. This decomposition results from observing the PDEs' mathematical structure. Linear parts have a convolutional representation through BINet, allowing time-independent training in advance. Nonlinear parts resemble CNN activation processes. A well-designed time discretization integrates structures into a NeuralNet architecture, resembling a CNN mechanism. Green's function training is a one-time effort for inference, evolution, and solving similar equations. Energy-stable property and a designed time discretization maintain energy, ensuring physical reliability in the proposed NeuralNet framework.
>
> 2.	Specifically, in the first step, we train the kernel function and the Green's function (if necessary), and in subsequent steps, all that is required is network inference and numerical integration calculations. The training of our Green's function is a one-time effort and can be used for the inference, evolution, and even the solution of other similar equations after the fact.
>
> 3.	Applying our method to more complex domains is theoretically feasible, but in practice, it may pose many challenges. This would be a long-term aspect of our work that needs further exploration.
>
> 4.	Regarding the concern about interpretability, we integrated mathematical knowledge into our method precisely because of the lack of interpretability in some neural network methods, aiming to enhance the interpretability of the equation solutions. Nevertheless, achieving complete interpretability in deep learning is a challenging area for us, as well.
>
> 5.	Yes, our method is more difficult than some other methods such as PINNs in coding, but our advantages has shown in 1.
>
> 6.	Please see 3.
>
> 7.	This may require extensive study including both sufficient computational experiments and deep theoretical thinking; this is indeed one of the long-term tasks in our future planning.
>
> 8.	In Appendix D.1, we compared the accuracy of the NEKM method with the PINN method; in Appendix D.2, we explained situations where the PINN method is challenging to train, while our method remains effective.
>
> Responses to Questions:
>
> 1.	Training the neural network in Equation 4 was very fast, sometimes taking less than a minute. Training the Green's function could take longer. This can be further improved if more sophisticated machine could be used (our computation is based on TITAN XP GPU). After training, calculations can be performed on a CPU, which usually does not take much time.
>
> 2.	The proposed method mainly rely on numerical integration, which we believe is not quite sophisticated for practitioners. In fact, given the advantages we elaborate in our response to Weakness 1, the CNN-like structure of our method makes it easily generalized.
>
> 3.	Please see 3 in ‘Weakness’ part.
>
> 4.	Please see 4 in ‘Weakness’ part.
>
> 5.	Energy-stable properties are derived from the physical features or inferred from the equation itself. Taking the Allen-Cahn equation as an example, it describes phase transitions in materials. The solution u represents the material's phase field, and the equation's evolution reflects changes in the phase field. In NEKM, the network not only outputs the solution's evolution but also ensures a certain energy decay.
>
> 6.	When handling singular boundary integrals, it is true that our method introduces some complexity at the code level. But it contributes to improved accuracy.
>
> 7.	Yes, it is a long-term challenge, and we need to invest more effort to address it.
>
> 8.	Please see 8 in ‘Weakness’ part.
>
> 9.	We believe that the tasks we can prioritize mainly involve solving problems in more complex domains, including how to improve the accuracy of the method and reduce computation time. In fact, we already have collaborators who have proposed using neural networks to approximate G_0, which will reduce the time required for directly computing G_0 using explicit expressions.
>
> 10.	It's evident that the reviewer values the practical utility of a method and its potential integration with engineering problems. This is an aspect we will pay attention to, and some of our collaborators are from commercial companies, so we will pay attention to their needs and continuously modify and improve our method.
>
> Once again, we thank the reviewer for his/her helpful suggestions and questions; his/her positive recognition will be crucial to our continued advancement!

---

### Meta-Review · Area_Chair_QEoz · 2023-12-10

**Metareview:**

The paper proposes a method for solving time-dependent PDEs by utilizing the knowledge of Green functions.

1) The approach potentially is applicable to interesting PDEs.

Weaknesses:
1) The paper is not well-written. It is very difficult to decipher the actual algorithm from the paper. What is trained? On what data? With what loss function? What is the complexity? How many parameters are trained? This is completely unreproducible, unclear and is just not ready for publication in any form.
2) Once the details are recollected from different parts of the paper (namely, the BiNet approach that learns the density) it becomes clear that the paper uses the operator splitting plus BiNet to learn the neural network approximation to boundary potentials. In this case, the number of baselines for the comparison should be much larger.

**Justification For Why Not Higher Score:**

The paper is not ready, it is not clearly written and does not contain minimal number of baselines.

**Justification For Why Not Lower Score:**

N/A

---

### Decision · Program_Chairs · 2024-01-16

Reject